# Interpol-IAGOS: a new method for assessing long-term chemistry-climate simulations in the UTLS based on IAGOS data, and its application to the MOCAGE CCMI-REFC1SD simulation

Yann Cohen[1,2,*], Virginie Marécal[1], Béatrice Josse[1], and Valérie Thouret[2]

[1]Centre National de Recherches Météorologiques, Université de Toulouse, Météo-France, CNRS, Toulouse, France

[2]Laboratoire d'Aérologie, University of Toulouse, CNRS, IRD, Université Toulouse 3 – Paul Sabatier (UPS), Toulouse, France

[*]Now at Laboratoire des Sciences du Climat et de l'Environnement, LSCE-IPSL (CEA-CNRS-UVSQ), Université Paris-Saclay, 91191 Gif-sur-Yvette, France

*Correspondence to:* yann.cohen@lsce.ipsl.fr

**Abstract.**

A wide variety of observation data sets are used to assess long-term simulations provided by chemistry-climate models (CCMs) and chemistry-transport models (CTMs). However, the upper troposphere–lower stratosphere (UTLS) has hardly been assessed in these modelling exercises yet. Observations performed in the framework of IAGOS (In-service Aircraft for a Global Observing System) combine the advantages of *in situ* airborne measurements in the UTLS with an almost global-scale sampling, a ∼20-year monitoring period and a high frequency. Even though a few model assessments have been made using the IAGOS database, none of them took advantage of the dense and high-resolution cruise data in their whole ensemble yet. The present study proposes a method to compare this large IAGOS data set to long-term simulations used for chemistry-climate studies. As a first application, the REF-C1SD simulation generated by the MOCAGE CTM in the framework of the CCMI phase-I has been evaluated during the 1994–2013 period for ozone ($O_3$) and the 2002–2013 period for carbon monoxide (CO). The concept of the new comparison software proposed here (so-called Interpol-IAGOS) is to project all IAGOS data onto the 3D grid of the model with a monthly resolution, since generally the 3D outputs provided by chemistry-climate models for multi-model comparisons on multi-decadal timescales are archived as monthly means. This provides a new IAGOS data set (IAGOS-DM) mapped onto the model's grid and time resolution. To get a model data set consistent with IAGOS-DM for the

comparison, a subset of the model's outputs is created (MOCAGE-M) by applying a mask that retains only the model data at the available IAGOS-DM grid points.

Climatologies are derived from the IAGOS-DM product, and good correlations are reported between with the MOCAGE-M spatial distributions. As an attempt to analyse MOCAGE-M behaviour in the upper troposphere (UT) and the lower stratosphere
(LS) separately, UT and LS data in IAGOS-DM were sorted according to potential vorticity. From this, we derived $O_3$ and CO seasonal cycles in eight regions well sampled by IAGOS flights in the northern mid-latitudes. They are remarkably well-reproduced by the model for lower-stratospheric $O_3$ and also good for upper-tropospheric CO.

Along this model evaluation, we also assess the differences caused by the use of a weighting function in the method when projecting the IAGOS data onto the model grid compared to the scores derived in a simplified way. We conclude that the
data projection onto the model's grid allows to filter out biases arising from either spatial or temporal resolution, and the use of a weighting function yields different results, here by enhancing the assessment scores. Beyond the MOCAGE REF-C1SD evaluation presented in this paper, the method could be used by CCMI models for individual assessments in the UTLS and for model intercomparisons with respect to the IAGOS data set.

## 1    Introduction

Chemistry-climate models (CCMs) and chemistry-transport models (CTMs) are essential tools for understanding atmospheric composition, for providing information where measurements are lacking, and for predicting air composition future evolution. Assessing and reducing uncertainties in the processes controlling its past and future changes can be achieved by comparing an ensemble of simulations from different models while using the same simulation setup. Among the model intercomparison projects, the main goal of the Chemistry-Climate Model Initiative (CCMI: Eyring et al., 2013) is the reduction of the uncer-
tainties in the multi-model projections involving stratospheric ozone, tropospheric composition and climate change, but also in a better understanding of the atmospheric processes relevant for these topics. CCMI is a common initiative from the International Global Atmospheric Chemistry (IGAC) and Stratosphere-to-troposphere Processes And their Role in Climate (SPARC) projects. It has taken over from both SPARC CCMVal (Chemistry-Climate Model Validation: SPARC, 2010) focused on the stratosphere and IGAC ACCMIP (Atmospheric Chemistry-Climate Model Intercomparison Project: Lamarque et al., 2013)
dealing mainly with tropospheric composition. In this framework, a set of simulations has been designed to address its objectives. Among them, the REF-C1SD experiment aims at assessing the ability of the models to reproduce the actual atmospheric composition for the recent climate time period. For this purpose, a part of its protocol consists of nudging the meteorological fields to meteorological reanalyses based on observations, as indicated by the SD suffix (which stands for "specified

dynamics"). The task for each participating model thus consisted of simulating as realistically as possible the tropospheric and stratospheric compositions in the last decades (1980–2010), following a common protocol.

Several studies have assessed the ability of REF-C1SD experiments, or previous similar simulations of air composition under recent climate conditions, to reproduce the mean tropospheric and/or stratospheric composition, by the use of monthly mean climatologies from observation data sets as reference, mostly from space. Froidevaux et al. (2019) based the evaluation of the REF-C1SD run from the CESM1-WACCM model on zonal monthly means of the stratospheric ozone column, using the Microwave Limb Sounder on Aura satellite (Aura-MLS) and the multi-satellite data set merged in the framework of the GOZCARDS (Global OZone Chemistry And Related trace gas Data records for the Stratosphere) project. As described in Young et al. (2018), tropospheric ozone fields provided by the ACCMIP participating models have been assessed, referring to zonally averaged mixing ratios from the Tropospheric Emission Spectrometer (Bowman et al., 2013), and tropospheric ozone column from OMI-MLS (Young et al., 2013). Hu et al. (2017) also compared the OMI-MLS tropospheric ozone columns to a GEOS-Chem simulation. The observed carbon monoxide (CO) columns from Measurement Of Pollution In The Troposphere (MOPITT) instrument served as the reference in the assessment of modeled tropospheric CO, notably from the REF-C1SD simulation generated by the GMI CTM over the period 2000–2010 (Strode et al., 2016), and from the Community Earth System Model (CESM1) CAM4-Chem model (Tilmes et al., 2016).

Only few studies compared observations (*in situ* measurements or from space) and CCMI REF-C1SD or similar simulations, focusing on the upper troposphere–lower stratosphere (UTLS). However, the latter is a key region regarding both the ozone ($O_3$) radiative forcing (Riese et al., 2012) and the stratosphere–troposphere exchanges (STEs) that substantially influence tropospheric ozone levels (e.g. Tao et al., 2019), albeit with a high uncertainty due to their different representations in models (Stevenson et al., 2006). Smalley et al. (2017) referred to the Aura-MLS measurements in the assessment of a 21[st] century projection (REF-C2) from twelve CCMs, focusing on their lower-stratospheric water vapour fields, during the 2004–2014 time period. *In situ* measurements with ozonesondes as part of the World Ozone and Ultraviolet radiation Data Center (WOUDC) have been compared to the REF-C1SD simulations from CMAM and EMAC models during the 2005–2010 time period (Williams et al., 2019). In addition to ozonesondes, aircraft measurements from different campaigns were used in the evaluation of the REF-C1SD simulations from the model CESM1 CAM4-Chem (Tilmes et al., 2016). Aircraft campaigns have already proven their usefulness in assessing models in the UTLS. Tilmes et al. (2010) built a climatology of $O_3$ and CO in the tropics, subtropics and extratropics by gathering a wide set of aircraft campaigns from 1995 until 2008. Hegglin et al. (2010) used this and other aircraft campaign-based data sets to assess the eighteen CCMs participating in CCMVal-2 in the extratropical lower stratosphere using several diagnostics. For instance, the seasonal cycles derived at 100 and 200 hPa highlighted

a relatively good reproduction of ozone behaviour in the lower stratosphere, and allowed to identify an overestimation of the transport from the tropics at 100 hPa and across the tropopause at 200 hPa. However, their conclusion also highlighted the limitations in space and time of the *in situ* observations, especially in the upper troposphere.

Among available observation data sets, the commercial aircraft measurements from the on-going IAGOS European Research Infrastructure (In-service Aircraft for a Global Observing System: Petzold et al., 2015, http://www.iagos.org) are well designed to study ozone and CO on the long term, notably in the UTLS (Cohen et al., 2018). IAGOS observations started in August 1994 for ozone and in December 2001 for CO. They are characterized by a high spatio-temporal resolution and a wide coverage with most data gathered at cruise levels (9–12 km above sea level). Thus, IAGOS database is suited to assess long-term simulations in this altitude range. Recently, its ozone data have been used to evaluate simulations from the models CESM1 CAM4-Chem (Tilmes et al., 2016) and GEOS-Chem (Hu et al., 2017) during the periods 1995–2010 and 2012–2013, respectively. Tilmes et al. (2016) used the IAGOS measurements gathered in the vicinity of Narita airport (Japan) only, and the comparison made by Hu et al. (2017) only spread over 2 years, while IAGOS ozone data are available since 1994 and covering a wide area, especially in the northern mid-latitudes from Western North America to East Asia. Brunner et al. (2003, 2005) combined research aircraft measurements with the first years of the IAGOS-MOZAIC database (1995–1998) to assess five CTMs and two CCMs. Gaudel et al. (2015) performed an evaluation of the MACC (Monitoring Atmospheric Composition and Climate) reanalysis over Europe during 2003–2010, using IAGOS $O_3$ and CO measurements. However, these comparisons used frequent simulation outputs. Although the high frequency is necessary for their approach to separate accurately the air masses into different categories, it is not adapted to the assessment of monthly averaged fields used in multi-model intercomparisons. Consequently, the IAGOS cruise data in the UTLS have been used neither as a whole ensemble nor to derive a monthly climatology for the evaluation of long-term chemistry-climate simulations. This is what we propose in the present paper.

To compare the REF-C1SD simulations against IAGOS data, interpolating the simulation outputs onto the high-resolution observations would be the most accurate way, but high-frequency outputs from multi-model intercomparisons such as CCMI are not available yet. Alternatively, the comparison could be performed after mapping the high resolution IAGOS data onto the model grid, on a monthly basis. Several gridding methods already exist for *in situ* measurements. Some of them consist of calculating a linear combination from the neighbouring measurements points onto each grid point (e.g. New et al., 2000). However, it requires to store simultaneously the information of all the measurement locations, and during a whole month. It is thus convenient for measurements with regular locations such as surface stations, whereas their use on the IAGOS database would be expensive computationally as well. Variational methods are also widely employed (e.g. Bourassa et al., 2005) but they concern data assimilation, which is not our purpose. The present study aims at providing a new methodology designed

to generate a gridded monthly data set from the IAGOS measurements, in order to evaluate REF-C1SD types of simulations. We also propose a set of relevant diagnostics for the model evaluation against IAGOS data mapped onto the model grid. These diagnostics originate from Cohen et al. (2018) who studied climatologies and trends in ozone and CO, based on the analysis of the the full IAGOS data set corresponding to the cruise phase of flights. The use of such a high spatial and temporal resolution data set allows to account for inter-regional differences that could not be highlighted with zonal means. Its projection onto a model grid suits well the constraint of working on monthly outputs from multi-decadal simulations like REF-C1SD. In order to demonstrate the interest of the new methodology and its associated diagnostics, we perform the assessment on one of the REF-C1SD simulations, that of the MOCAGE CTM.

In Sect. 2, we describe briefly the IAGOS observations, the CCMI model intercomparison project, the MOCAGE CTM that we use in this study, and its configuration for the REF-C1SD simulation. In Sect. 3, we present the methodology proposed to map the IAGOS data set onto the model grid on a monthly resolution, the chosen statistical metrics for models' evaluation and the different assessment diagnostics. In Sect. 4, we present a first application of this methodology on the evaluation of the MOCAGE REF-C1SD simulation. Strengths and weaknesses of the methodology and the chosen diagnostics are discussed. Conclusions are given in Sect. 5.

## 2  Observations and simulation

### 2.1  The IAGOS observations

The European Research Infrastructure IAGOS (Petzold et al., 2015, http://www.iagos.org) provides *in situ* measurements on board several commercial aircraft. The observations used hereafter have been performed in the framework of the on-going IAGOS-Core program that followed the MOZAIC program (Marenco et al., 1998). Ozone (resp. CO) measurements started in August 1994 (resp. December 2001), based on an UV (resp. IR) absorption technology, with an accuracy of 2 ppb (resp. 5 ppb), a precision of 2% (resp. 5%) and a time resolution of 4 s (resp. 30 s). Further information about the instruments can be found in Marenco et al. (1998) and Thouret et al. (1998) for $O_3$, and in Nédélec et al. (2003) for CO. Nédélec et al. (2015) present a more recent evaluation of both ozone and CO instruments in the frame of IAGOS.

The IAGOS observations (referring to the IAGOS-Core database hereafter) sample frequently the whole troposphere nearby airports, measuring vertical profiles during ascent and descent phases, and the UTLS during the cruise phases, mostly in the northern mid-latitudes where most of the flight observations are gathered. In these latitudes, a recent analysis of $O_3$ and CO climatologies and trends based on almost two decades of IAGOS cruise measurements has been performed in Cohen et al.

(2018). In addition to global climatologies, the same analysis also focused on eight well-sampled regions in the UT and the LS separately. In order to generate results comparable with the latter, this study focuses on the same time period (1994–2013) and, where relevant, on the same regions.

## 2.2 The CCMI project and the REF-C1SD experiment

5   The CCMI project is a common initiative from the IGAC and SPARC programs. The CCMI phase-1 gathers a community of 18 chemistry-climate models (CCMs) and two CTMs, which description is given in the review of Morgenstern et al. (2017). A series of experiments have been designed to model tropospheric and stratospheric air compositions for past, present and future climates. For each experiment, a common protocol is recommended to all participating models. Amongst the CCMI simulations, the REF-C1SD reference experiment aims at modelling as realistically as possible the day-to-day tropospheric 10  and stratospheric compositions in a recent climate, using specified dynamics (SD). For this purpose, as described in Eyring et al. (2013), the simulations are driven by (or nudged towards) dynamical reanalyses data sets (typically ERA-Interim or MERRA), and extending from 1980 until 2010. For this long-term simulation, the 3D outputs fields of species concentrations are archived as monthly means.

## 2.3 The MOCAGE model and the simulation set-up

15   The MOCAGE model (MOdèle de Chimie Atmosphérique à Grande Echelle: Josse et al., 2004; Guth et al., 2016) is an offline global chemistry-transport model (CTM). The chemical scheme is composed by the coupling of the RACM (Regional Atmospheric Chemistry Mechanism: Stockwell et al., 1997) and the REPROBUS (REactive Processing Ruling the Ozone BUdget in the Stratosphere: Lefèvre et al., 1994) schemes, corresponding to tropospheric and stratospheric chemistry, respectively. The MOCAGE REF-C1SD simulation is run using a global domain at a $2° \times 2°$ horizontal resolution, and 47 vertical levels, in 20  $\sigma$-hybrid pressure, distributed from the surface up to $\sim$5 hPa. The simulation is driven by the meteorological fields from the ERA-Interim reanalysis. The biomass burning and anthropogenic emissions come from the GFEDv2 and MACCity inventories, respectively. The latter is characterized by a 10-year resolution and a linear interpolation is applied to derive yearly emissions. The period spreads from August 1994 until December 2013, consistently with Cohen et al. (2018). The first 14 years come from the MOCAGE REF-C1SD simulation originally produced for CCMI project. For the years out of the period covered by 25  the experiment, the MOCAGE REF-C1SD run has been extended until December 31, 2013 using the same code and inputs as in the original MOCAGE CCMI REF-C1SD simulation.

## 3  Methodology

The objective of the proposed methodology is to make possible the comparison between the whole IAGOS database and the 3D monthly mean volume mixing ratios from CTMs and CCMs simulations. Our approach consists of distributing the IAGOS observations, performed every 4 s, on a given model grid. A first application is proposed on the MOCAGE REF-C1SD run, characterized by a ~200 km horizontal resolution in the mid-latitudes, and a ~800 m vertical resolution in the UTLS. In order to account for the distance of the measurements from the center within one given cell, we chose a reverse linear interpolation at the first order, as described in Sect. 3.1 and illustrated in Fig. 1. The subsequent gridded monthly means are derived using weighted averages, as described in Sect. 3.2, and are directly comparable to the model monthly mean outputs.

In a first step, this approach is used for a statistical evaluation of the MOCAGE REF-C1SD climatologies on a hemispheric scale over the periods December 1994–November 2013 for $O_3$ and December 2001–November 2013 for CO. The data processing used to produce the climatologies and the statistical metrics chosen are presented in Sect. 3.3. In a second step, we attempt to go further in the assessment of the MOCAGE simulation by evaluating separately the upper troposphere and the lower stratosphere. For this purpose, the discrimination between the grid points mostly representative of the UT or the LS is necessary. As in Cohen et al. (2018), this has been done with respect to Ertel potential vorticity (PV) and applied in eight northern mid-latitude regions selected because of their high level of sampling by IAGOS. The methodology used is explained in Sect. 3.4.

### 3.1  Reverse interpolation of a given measurement point on the model grid

At a given point where IAGOS measured a mixing ratio $C_{obs}(X)$ for species X, the algorithm presented here locates its position on the model grid defined by its longitude, latitude and $\sigma$-hybrid pressure coordinates. More precisely, we locate the model grid point which is the closest west and south of, and below (in altitude) the observation point and which corresponds to the $i^{th}$, $j^{th}$ and $k^{th}$ grid point coordinates respectively. As shown in Fig. 1c, a normalized scalar is then computed for each dimension (coefficients $\alpha$, $\beta$, $\gamma$), increasing linearly with the distance between the measurement point and the (i, j, k) grid point. Note that the $\gamma$ vertical coefficient is derived from log-pressure coordinates. Finally, a resulting 3D weight is computed for each of the eight closest cells. By noting the variable indexes I, J and K belonging to the ensembles {i, i+1}, {j, j+1} and {k, k+1} respectively, we define the functions $f_I$, $g_J$ and $h_K$ which values depend on $\alpha$, $\beta$ and $\gamma$ respectively, such as:

$$f_I(\alpha) = \begin{cases} 1-\alpha & \text{if } I=i \\ \alpha & \text{if } I=i+1 \end{cases} \quad ; \quad g_J(\beta) = \begin{cases} 1-\beta & \text{if } J=j \\ \beta & \text{if } J=j+1 \end{cases} \quad ; \quad h_K(\gamma) = \begin{cases} 1-\gamma & \text{if } K=k \\ \gamma & \text{if } K=k+1 \end{cases} \tag{1}$$

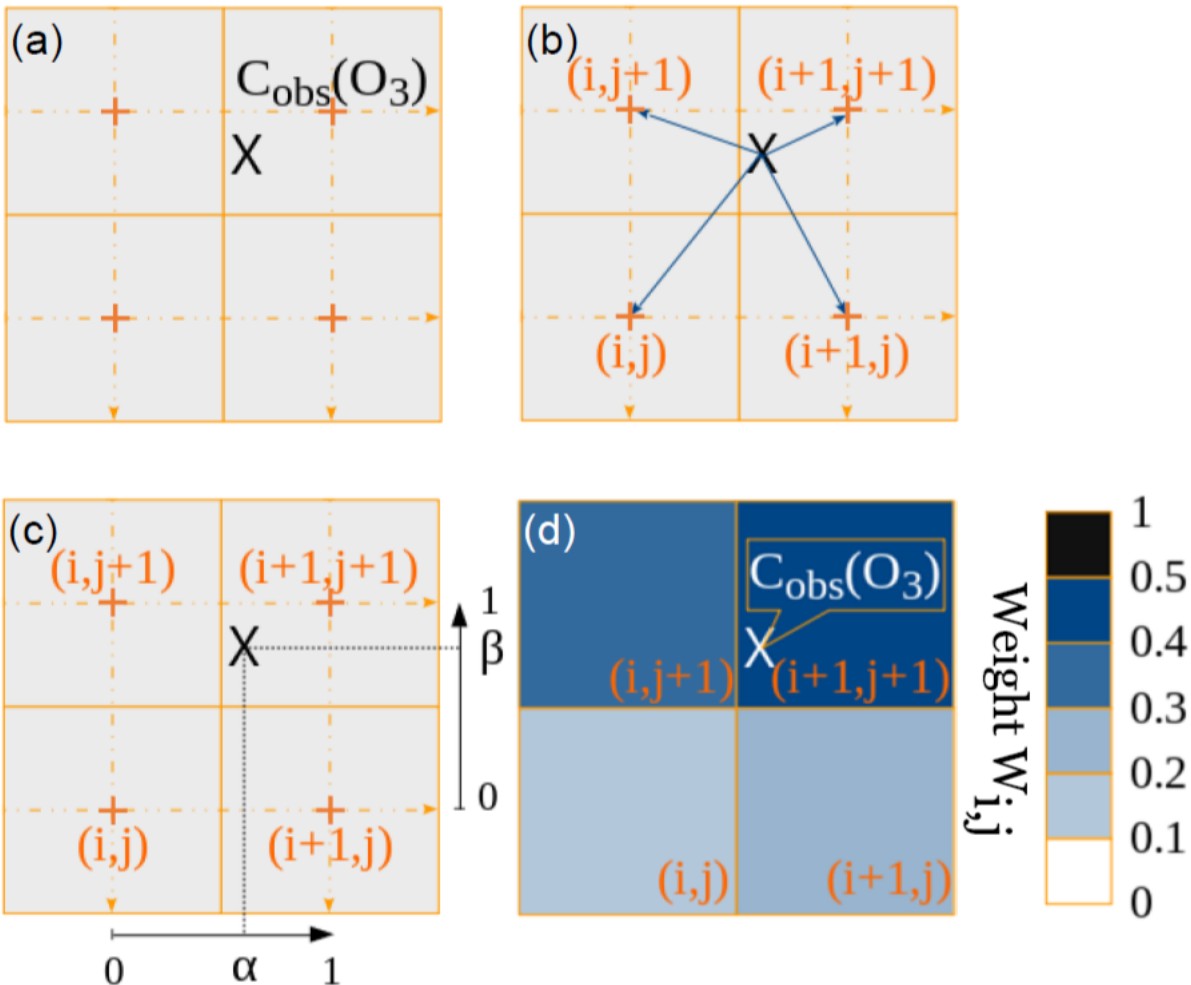

**Figure 1.** Schematic of the method used for the distribution of the observations on the model grid, represented in two dimensions for simplicity. (a) shows the location of a chosen measurement point in the model's grid. The steps of the method are (b) to locate the model's grid points (orange crosses) closest to this location; (c) to calculate a normalized scalar for each dimension ($\alpha$ and $\beta$), depending on the distance between the measurement point and the "bottom-left" grid point; (d) the calculation of the weight for the four closest grid points. As indicated in the colour scale on the right, this weight ranges between 0 and 1.

The resulting weight for each of the grid points surrounding the measurement location is thus defined as the following product:

$$W_{I,J,K}(\alpha,\beta,\gamma) = f_I(\alpha).g_J(\beta).h_K(\gamma) \tag{2}$$

In this way, as illustrated in Fig. 1d, for a given cell (I, J, K) amongst the eight closest ones, this weight decreases when the distance increases between the measurement point and the model grid point. Note that since the simulation outputs are monthly averages, we use the monthly mean surface pressure for determining the hybrid $\sigma$-pressure on the 47 vertical grid levels for a given model longitude/latitude. Although the surface pressure can show an important intra-monthly variability, we calculated that a 30 hPa change at surface would cause a variation weaker than 2 hPa on a given vertical grid level in the UTLS. Although caution is needed while treating low-altitude measurements, the monthly resolution on the surface pressure field thus has a negligible impact on the distribution of the IAGOS data from the cruise altitudes onto the model vertical grid.

### 3.2  Deriving the monthly mean values from observations

The weighting coefficients defined above correspond to one single observation data point. To obtain monthly averages from the whole observation data set, the last step consists of summing up all the values measured in the vicinity of the (i, j, k) grid point for each month. Thus, for a given grid point (i, j, k), we define n as the index for the measurement performed in its vicinity during the considered month, and the corresponding mixing ratio for the species X is noted $C_{obs, n}(X)$, and N the total number of measurements performed in its vicinity. The monthly value of the X mixing ratio at (i, j, k) is then derived with the equation:

$$X_{i,j,k} = \frac{\sum_{n=1}^{N} W_{i,j,k,n}(\alpha,\beta,\gamma)C_{obs,n}(X)}{\sum_{n=1}^{N} W_{i,j,k,n}(\alpha,\beta,\gamma)} \tag{3}$$

where the denominator is equivalent to the amount of weighted measurement points performed in the (i, j, k) grid cell during the chosen month. Hereafter, we refer it as $N_{eq}$.

In the end, this method yields monthly fields of IAGOS $O_3$ and CO mixing ratios (or any other variable measured by IAGOS, e.g. water vapour) projected on the MOCAGE grid points where IAGOS data are available. This data set is named IAGOS-DM hereafter, the suffix DM referring to the distribution on the model grid. With this method, the cruise observation data are distributed onto the MOCAGE vertical levels spanning from level 28 up to level 22 and corresponding to the ~360–175 hPa interval. Note that the measurement points on the MOCAGE vertical levels below level 28 (~360 hPa) are considered as

corresponding to ascent or descent phases of the flights. These measurements are not processed, since they are only available in small areas close to airports. Levels 27 and 28 generally correspond to these phases too but include cruise measurements above elevated lands, since hybrid sigma-pressure levels tend to follow land elevation. In order to compare the observations and the model at the same locations and months, we apply a mask on the MOCAGE REF-C1SD simulation outputs that allows us to

account only for the IAGOS-DM sampled grid points. The subsequent data set is named MOCAGE-M, the M letter referring to the mask. Thus, IAGOS-DM and MOCAGE-M data sets are spatially consistent and can be used to make gridpoint-to-gridpoint comparisons on climatological timescales, as long as we assume the gridded IAGOS data to be representative of the measurement period. The latter point has been tested on a 5-year subsample using the simulation daily outputs instead of monthly outputs. It consisted of comparing MOCAGE-M to a test product derived by calculating monthly averages from the

daily outputs and applying a mask based on the IAGOS daily sampling. The results from this test are briefly presented in Sect. 4.

In order to test the advantages of the linear interpolation involving weighting factors, we also derive another product from the IAGOS database using a simplified method, i.e. by solely averaging the measurement points into the grid cells where they are located. This control product is named IAGOS-DM-noW hereafter, the -noW suffix standing for "no weighting". Since it

changes the spatial sampling distribution, a new subsequent mask has to be applied to the MOCAGE grid to be consistent with the IAGOS-DM-noW product, so-called MOCAGE-M-noW hereafter.

### 3.3    Methodology for the assessment of the climatologies

#### 3.3.1    Filtering conditions

For the climatological part of this study, we chose to perform a seasonal and a yearly analysis. Avoiding sampling biases where

and when IAGOS-DM data (counted as $N_{eq}$) are not numerous enough requires that the seasonal sample $N_{eq}$ reaches a minimum threshold to be selected (noted $N_{thres}$). We chose to set this $N_{thres}$ limit depending on latitude to account for the varying gridbox area linked to the $2° \times 2°$ grid, and on the chemical tracer to account for the shorter period for CO measurements compared to $O_3$. $N_{thres}$ therefore decreases with latitude following a cosine function, similarly to the model horizontal grid cell areas. The reference threshold $N_{thres, ref}$ corresponds to $O_3$ measurements for gridbox areas during a given season, over the whole period

(December 1994–November 2013 for $O_3$ and December 2001–November 2013 for CO). It has been set to $N_{thres, ref} = 100$ as a compromise between sampling robustness and a large-enough amount of data in IAGOS-DM sample. Accounting for the shorter CO measurement period compared to $O_3$ ($\sim 60\%$ of the $O_3$ period), the same threshold applied to the CO climatologies would result in a greater proportion of filtered-out grid cells. Thus, the corresponding $N_{thres}$ threshold for this species is derived

by applying a factor 0.6, leading to 60. Note that this reference threshold is defined seasonally. Therefore, the $N_{thres, ref}$ used for yearly climatologies is multiplied by a factor 4.

### 3.3.2 Statistical metrics for assessing the climatologies

Quantifying a simulation assessment requires the use of statistical parameters. This paragraph aims at defining the chosen metrics, and at justifying this choice. Pearson's coefficient is a key result from linear regressions. It is used to quantify the correlation between two signals. If we call $(m_i)_{i \in [\![1,N]\!]}$ and $(o_i)_{i \in [\![1,N]\!]}$ the lists of modelled and observed values respectively, their correlation is defined as:

$$r = \frac{1}{N} \frac{\sum_{i=1}^{N}(m_i - \bar{m})(o_i - \bar{o})}{\sigma_m \sigma_o} \tag{4}$$

where $\bar{m}$ and $\bar{o}$ are the mean values and $\sigma_m$ and $\sigma_o$ their respective standard deviations. Quantifying total biases and mean errors is also primordial in a model assessment. However, the use of the absolute mean bias and root mean square error (RMSE) may not be relevant for climatological purposes because of a strong influence that could arise from from observed outliers. In our context, another inconvenience lies in the strong vertical $O_3$ gradient near and above the tropopause. It tends to induce a strong absolute bias with respect to the tropospheric mixing ratios, since it makes the $O_3$ absolute mean bias and RMSE mainly depending on the highest vertical grid cells. The normalized bias metric (and associated standard error) is chosen for a better representativeness of biases for both low and high mixing ratios. The modified normalized mean bias (MNMB) and the fractional gross error (FGE) are respectively defined as:

$$MNMB = \frac{2}{N} \sum_{i=1}^{N} \frac{m_i - o_i}{m_i + o_i} \tag{5}$$

and

$$FGE = \frac{2}{N} \sum_{i=1}^{N} \left| \frac{m_i - o_i}{m_i + o_i} \right| \tag{6}$$

The MNMB (resp. FGE) represents precisely the spatial average based on the model relative biases (resp. on their absolute value) shown in Figs. 3, 4 and in Sect. A in Appendix.

### 3.4 Methodology for assessing the seasonal cycles in the UT and in the LS

A second part of this assessment targets the behaviour of the model in the UT and the LS separately. The diagnostics we use for this purpose are adapted from Cohen et al. (2018), based on Thouret et al. (2006) who used the potential vorticity (PV) fields from the ECMWF operational analysis to derive the tropopause pressures. In contrast with the latter studies, we define the tropopause layer with the monthly averaged PV fields from ERA-Interim, as used in the MOCAGE REF-C1SD simulation. A given grid point is considered as belonging to the UT if its monthly PV is lower than 2 potential vorticity units (PVU), and to the LS if the PV is greater than 3 PVU. The cells which PV ranges between 2 and 3 PVU are considered as belonging to the transition zone separating the two layers and are not selected. In order to enhance the distinction between the UT and the transition zone, the first model level below the 2 PVU threshold is also filtered out from the UT. The 2 PVU threshold is derived from a log-pressure interpolation between the grid points. We also filter out the grid boxes where this PV classification is not consistent with the mean observed $O_3$ mixing ratio, i.e. where the monthly $O_3$ level reaches 140 ppb in the UT and where it goes under 60 ppb in the LS. It avoids an additional bias based on errors in the dynamical field leading to unrealistic UT and LS attribution. These thresholds on $O_3$ mixing ratio were chosen according to the $O_3$ seasonal cycles shown in Fig. 3.7 in Cohen et al. (2018), where the upper boundary linked to the interannual standard deviation in the UT is less than 100 ppb and where the lower boundary in the LS is greater than 100 ppb. We estimated that a supplementary 40 ppb interval would limit an exaggerated filtering of grid cells monthly values.

As in Cohen et al. (2018), we focus our analysis on the seasonal cycles for eight regions in the northern mid-latitudes that are well sampled by IAGOS. Their coordinates and their corresponding sampling are detailed in Table 1 in Cohen et al. (2018). Because of the $2° \times 2°$ horizontal grid resolution in the simulation, we applied a $1°$ eastward or northward shift on the odd-coordinated edges. The subsequent regions defined in this paper are shown in Fig. 2. For each of them, the monthly means are calculated by averaging the gridded monthly means separately in the UT and the LS. The latter values were defined as described in Sect. 3.1 and 3.2.

In Cohen et al. (2018), the regional monthly means with less than 300 data were filtered out. Here, due to the loss of data caused by the monthly resolution, we lowered this minimum threshold to 150 in order to keep taking the less sampled regions into account, such as Western North America and Siberia. Still, we kept the criterion from Cohen et al. (2018) which required at least 7 days between the first and last measurements in the considered month and region, avoiding the averages to be representative of transient meteorological conditions only. Following the same study, the computation of the seasonal cycles is based on the years exhibiting seven available months or more, distributed on three seasons at least. This criterion avoids biases linked to the inter-seasonal differences in the sampling, thus ensuring a good representativeness of the whole year. It is important to note that

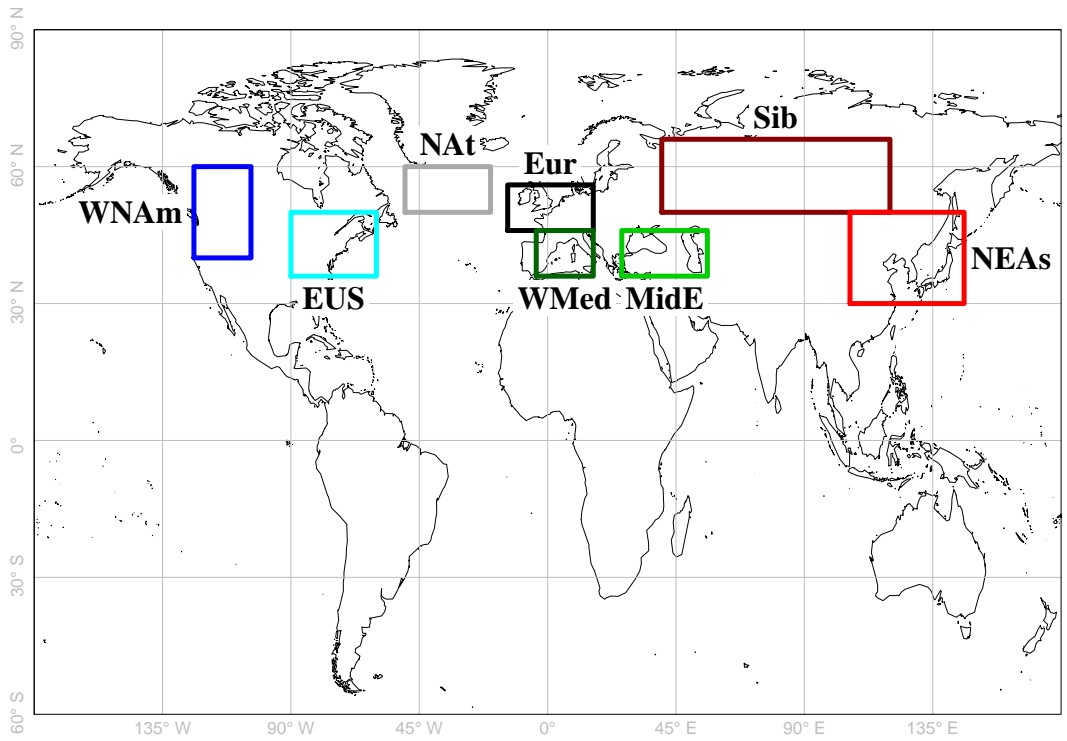

**Figure 2.** Map of the regions selected for this study based on Cohen et al. (2018), adapted to MOCAGE global grid. From West to East, the acronyms stand for Western North America, Eastern United-States, Northern Atlantic, western Europe, West Mediterranean basin, Middle East, central Siberia and Northeast Asia.

the sampling threshold mentioned in this paragraph concerns each monthly average within a regional time series, contrasting with the sampling threshold we use for the (multi-)decadal average on each grid cell in the horizontal climatologies.

## 4 Results

### 4.1 Monthly representativeness

5 A first step in the assessment of the methodology consists of testing the monthly representativeness of the IAGOS-DM mean values, in order to evaluate the temporal consistency between IAGOS-DM and MOCAGE-M. For this purpose, as mentioned at the end of Section 3.2, we compared MOCAGE-M to a test product derived by calculating monthly averages from the simulation daily outputs, after applying a mask based on the IAGOS daily sampling. For this test, the chosen period spreads from 2003 until 2007 included, an uninterrupted measurement period for both ozone and CO. Concerning the mean 3D distribu-

tions, a mean normalized difference between the two products has been found below 1.7 % for each season and each species. In absolute values, 10 % of the yearly mean biases are greater than 6.0 % (4.1 %) for ozone (CO), and 1 % greater than 13.1 % (10.3 %). Seasonal mean biases are characterized by a 90 percentile generally lower than 10 %, and a 99 percentile from 14.7 up to 22.8 % for ozone and from 13.0 up to 18.1 % for CO. The maximum values correspond to winter and spring. Concerning

the seasonal cycles, the relative difference between the two MOCAGE products was found to be almost systematically below 5 %, and amongst all the regions, its ozone values seldom outreach 10 %, with a maximum value at 15.2 %. In conclusion of this comparison, the similar results obtained between MOCAGE-M and the test product suggested that in most cases, the IAGOS-DM monthly means could be considered as representative of the month.

## 4.2   Horizontal climatologies

Figures 3 and 4 show the yearly mean climatologies, respectively for $O_3$ and CO, and the model relative biases. The latter are defined as the model bias normalized to the average between the two data sets, and are provided in percentages in these figures. The level 22 is seldom reached by the IAGOS measurements, and the levels 27 and 28 are sampled only in the vicinity of airports. Thus, only the levels 26 up to 23 are represented in these figures. Additionally, the seasonal mean climatologies are available in Appendix A.

In Fig. 3, IAGOS-DM and MOCAGE-M show similar geographical structures. In the tropics and subtropics, the $O_3$ amounts are close, with consistent poleward gradients. Both have maxima located above Northeast Canada. The $O_3$ mixing ratio in the northern mid-latitudes is underestimated in the model for the levels 24–26, and close to the observations for the level 23. The seasonal climatologies in Figs. A1–A4 show that this feature is representative of spring and fall, whereas ozone tends to be underestimated (resp. overestimated) in all vertical grid levels in summer (resp. winter). Note that the discontinuity over

Greenland is due to its topography causing a steep elevation of the vertical grid levels.

In Fig. 4, CO also shows a good correlation between the two data sets, notably with the same maxima and minima locations. But the CO mixing ratio is generally overestimated by the model, especially over East Asia and India. In the northern mid-latitudes, the seasonal climatologies in Figs. A5–A8 generally show an overestimation in winter and spring and a less-visible underestimation in summer and fall.

Figure 5 proposes a synthesis for the comparison between the yearly climatologies over the whole period. The same figures can also be found for each season in Appendix B. The linear regression parameters indicated in the graphs show a strong geographical correlation, its coefficient spreading from 0.73 up to 0.97. The correlation is better for $O_3$ (>= 0.92), and at higher levels for both species. Consequently, the geographical distributions in $O_3$ and CO are well reproduced in the simulation. Their

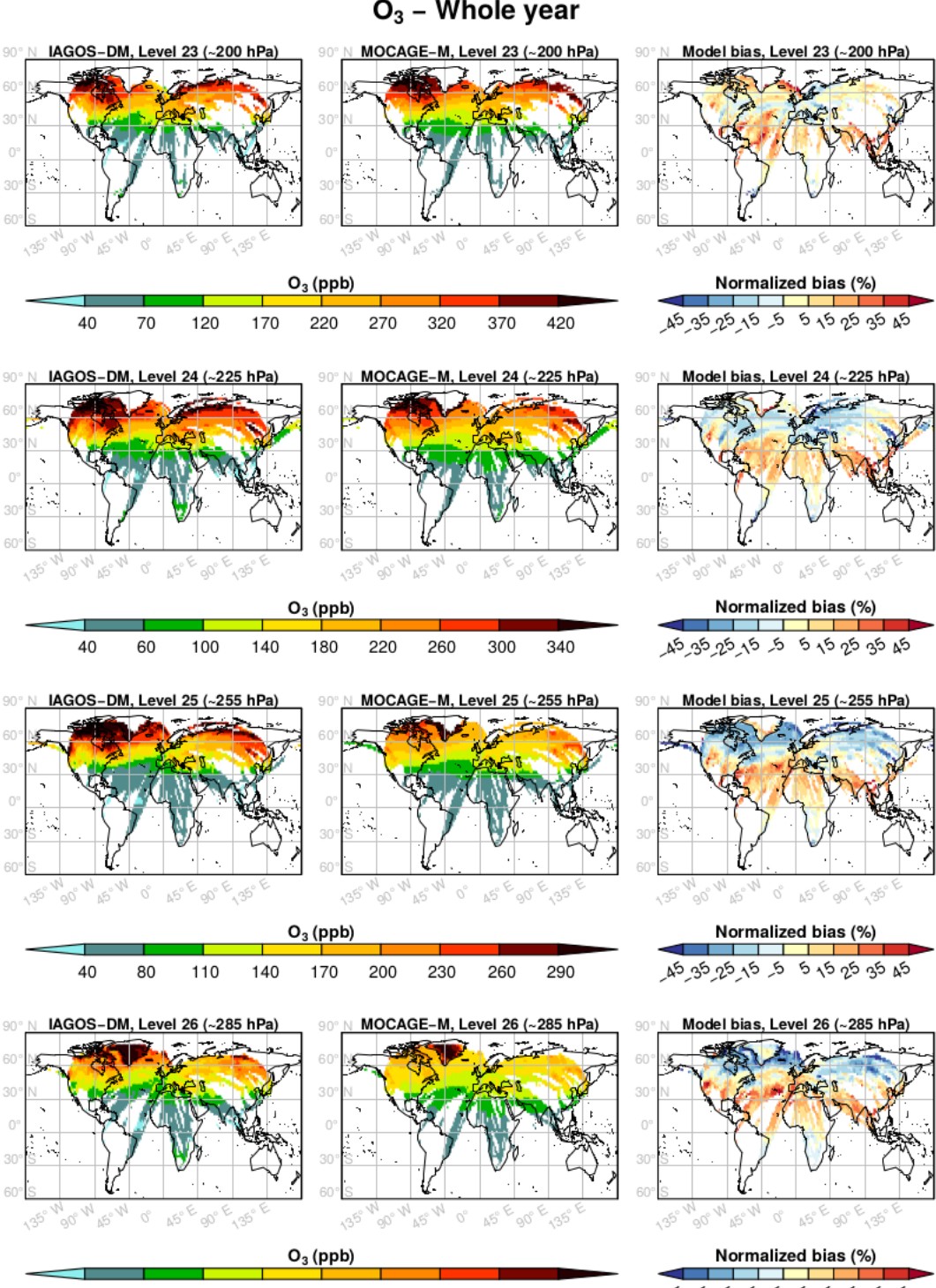

**Figure 3.** Mean horizontal distribution of the O₃ mixing ratio (ppb) from the model levels 26 (∼320 hPa, bottom panels) to 23 (∼200 hPa, top panels) in IAGOS-DM (left panels) and MOCAGE-M (middle panels) during the period December 1994–November 2013. The normalized bias is represented in the right panels, in percentage with respect to the average between the two data sets.

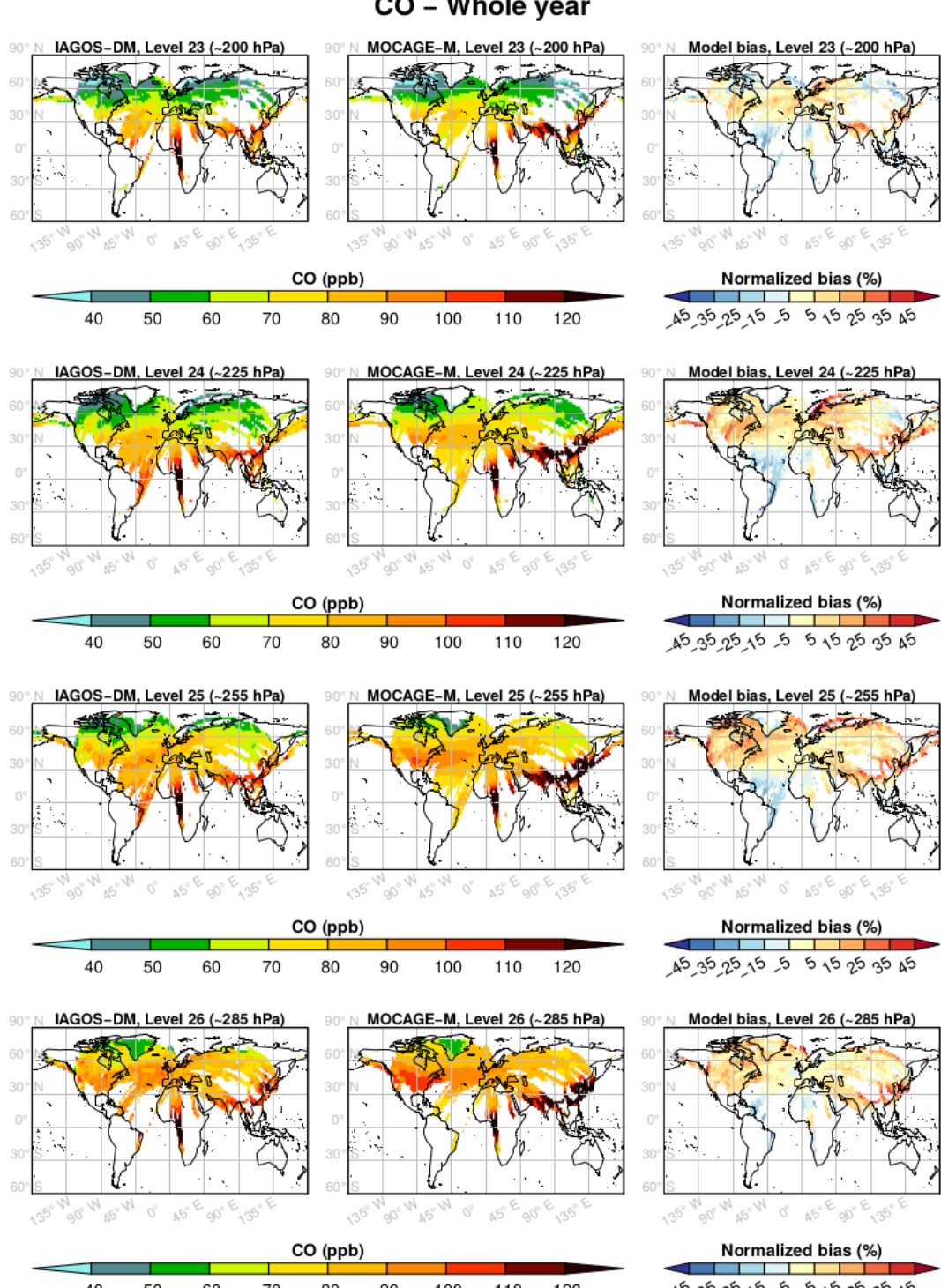

**Figure 4.** As in Fig. 3 for CO, during the period December 2001–November 2013.

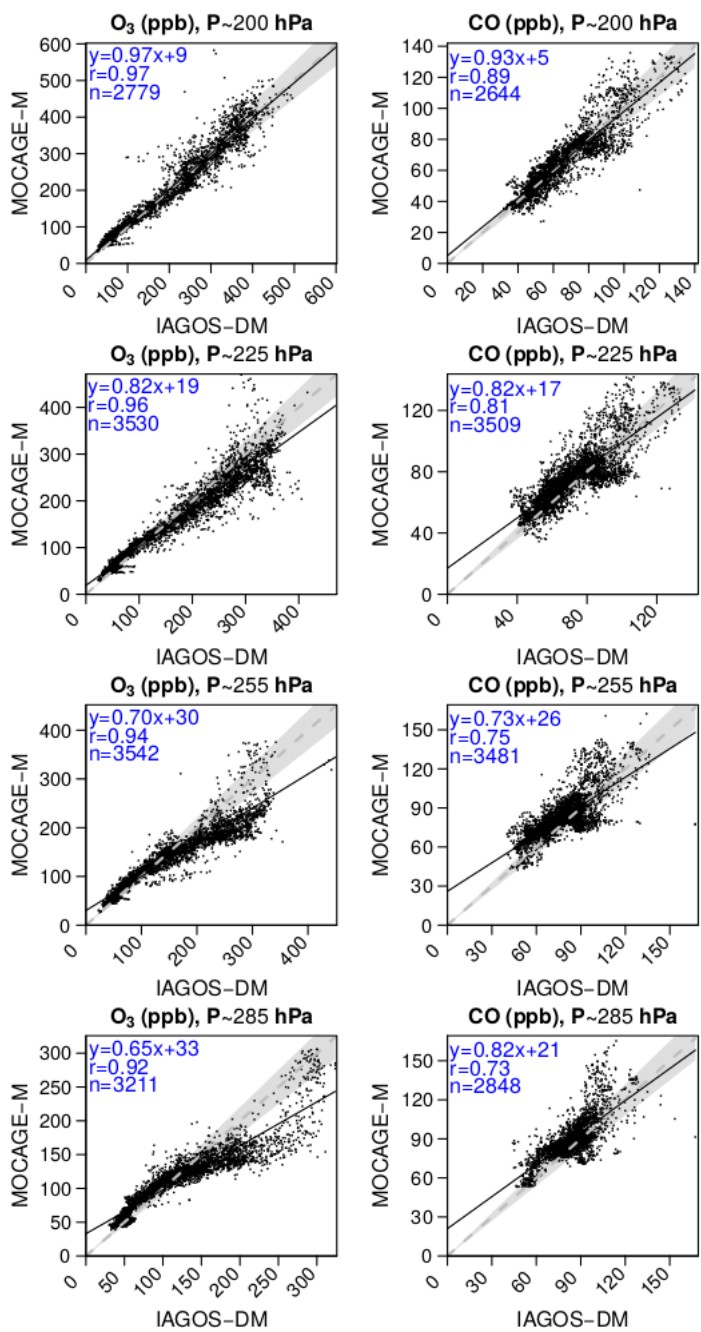

**Figure 5.** Scatterplots comparing the yearly mixing ratios from MOCAGE-M and IAGOS-DM, for $O_3$ (left panels) and CO (right panels), at each vertical grid level. The linear regression fit is represented by the solid black line. The dashed gray line represents the $y = x$ reference line, and the shaded area corresponds to a 10% error. The regression coefficients and the amount of data (n) are written in the top-left corner of each panel.

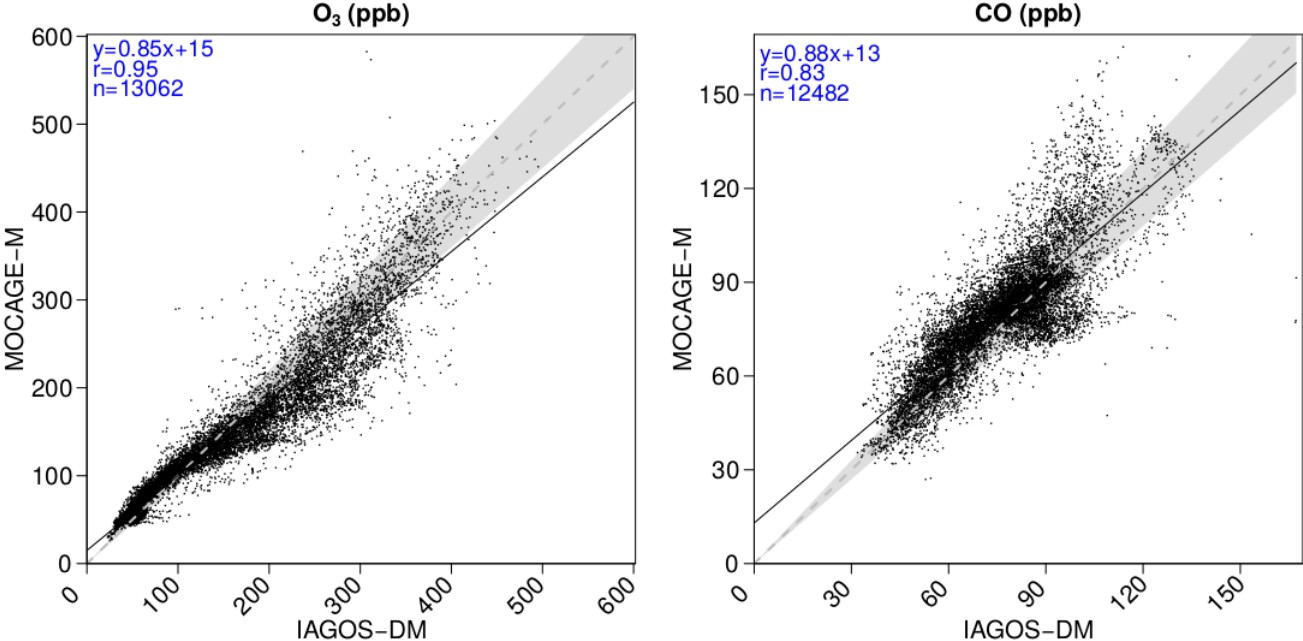

**Figure 6.** Same as Fig. 5, gathering the points from the four vertical levels.

stronger correlations at higher levels suggest a remarkably good consistence of the modelled stratospheric composition with the observations, showing its ability to simulate stratospheric chemistry and transport. The same feature is visible with the regression fit, showing a lower bias for $O_3$, and at highest levels. With respect to the 1:1 line, levels 25 and 26 are characterized by an overestimation of the lower part of the $O_3$ distribution (< 120 ppb) and by an underestimation of the higher part, more

pronounced during boreal summer according to Fig. B3. A possible reason is that the summertime tropopause altitude in these regions can be overestimated by the model, or that the vertical stability is underestimated. These biases have been largely improved with the most recent version of MOCAGE used to run CCMI phase-2 simulations. Concerning CO, the highest values (generally > 100 ppb) correspond to the strongly emitting and convective regions: South Asia, East Asia and tropical Africa. A supplementary test (not shown here) has identified the high mixing ratios close to the 1:1 line at tropical African

points, whereas the high mixing ratios with a positive bias were associated both to South and East Asia areas. The latter can be due to an overestimation of convection in this region and/or an overestimation in the inventory for Asian emissions. On the contrary, CO above tropical Africa shows good results, indicating a realistic combination between convection and emissions.

     The method proposed in this paper to evaluate MOCAGE REF-C1SD against IAGOS data in the UTLS aims at being applied to other chemistry-climate simulations, like the REF-C1SD simulations from other models. Since IAGOS is mapped onto the

model vertical grid, the latter differing from one model to another, we also plotted a synthetic regression in Fig. 6, where

**Table 1.** Seasonal and annual metrics synthesizing the assessment of the simulated $O_3$ and CO climatologies by IAGOS-DM, gathering all the vertical grid levels as in Fig. 6. From left to right: Pearson's correlation coefficient (r), modified normalized mean bias (MNMB), fractional gross error (FGE) and the sample size ($N_{cells}$).

| Species | Season | r | MNMB | FGE | $N_{cells}$ |
|---------|--------|------|--------|-------|---------|
| $O_3$ | DJF | 0.95 | 0.144 | 0.190 | 12,723 |
| | MAM | 0.94 | -0.033 | 0.163 | 12,622 |
| | JJA | 0.92 | -0.169 | 0.280 | 12,587 |
| | SON | 0.89 | 0.027 | 0.180 | 13,073 |
| | ANN | 0.95 | -0.012 | 0.150 | 13,062 |
| CO | DJF | 0.77 | 0.098 | 0.171 | 12,081 |
| | MAM | 0.82 | 0.098 | 0.157 | 11,623 |
| | JJA | 0.75 | -0.011 | 0.130 | 11,618 |
| | SON | 0.75 | 0.024 | 0.126 | 12,467 |
| | ANN | 0.83 | 0.049 | 0.112 | 12,482 |

**Table 2.** Same metrics as in Table 1, showing the scores derived from the comparison between IAGOS-DM and MOCAGE-M (first three columns), then between IAGOS-DM-noW and MOCAGE-M-noW. All the scores reported in this table are based on the IAGOS-DM-noW sampling size $N_{cells, noW}$ (last column).

| Species | Season | r | MNMB | FGE | $r_{noW}$ | $MNMB_{noW}$ | $FGE_{noW}$ | $N_{cells, noW}$ |
|---------|--------|------|--------|-------|------|--------|-------|---------|
| $O_3$ | DJF | 0.95 | 0.133 | 0.182 | 0.91 | 0.076 | 0.219 | 10,404 |
| | MAM | 0.95 | -0.042 | 0.165 | 0.91 | -0.101 | 0.260 | 10,293 |
| | JJA | 0.92 | -0.190 | 0.287 | 0.88 | -0.217 | 0.354 | 10,252 |
| | SON | 0.90 | 0.009 | 0.177 | 0.84 | -0.021 | 0.232 | 10,684 |
| | ANN | 0.95 | -0.021 | 0.153 | 0.92 | -0.073 | 0.224 | 11,667 |
| CO | DJF | 0.74 | 0.120 | 0.173 | 0.67 | 0.182 | 0.243 | 9,298 |
| | MAM | 0.79 | 0.105 | 0.156 | 0.72 | 0.186 | 0.247 | 8,896 |
| | JJA | 0.74 | -0.013 | 0.122 | 0.63 | 0.042 | 0.168 | 8,977 |
| | SON | 0.72 | 0.035 | 0.121 | 0.68 | 0.077 | 0.166 | 9,608 |
| | ANN | 0.81 | 0.056 | 0.110 | 0.71 | 0.113 | 0.178 | 10,735 |

all the points at all levels have been gathered into one single scatterplot. These summarized model performances concerning mean spatial distributions are the final products of our evaluation methodology for climatologies. From the whole ensemble of ~13,000 (~12,500) sampled grid points for $O_3$ (CO), the correlation shows a good agreement between the simulation and the observations, especially for $O_3$ (r=0.95). Its regression fit is dominated by an overestimation for lower values (< 100 ppb) and

5    an underestimation for higher values, especially between 200 and 300 ppb. Above 350 ppb, the balance between overestimated and underestimated $O_3$ values tends to be more balanced.

Table 1 gives a synthesis of the biases and associated deviations, for the assessment of MOCAGE-M versus IAGOS-DM. The yearly MNMB equals -0.012 for $O_3$ and 0.049 for CO, demonstrating a very good estimation of these two species in the UTLS on a hemispheric scale, especially for $O_3$. More precisely, it shows a balance between positive and negative normalized

10   biases. The yearly fractional gross error (FGE), corresponding to the averaged normalized bias absolute value is also low, with

0.150 and 0.112 for $O_3$ and CO respectively. The seasonal patterns show that metrics linked with CO biases (MNMB and FGE) generally yield values closer to 0, compared to $O_3$. The $O_3$ seasonal behaviour is characterized by a balance between opposite seasons: the most positive (resp. negative) bias takes place in winter (resp. summer) and equals 0.144 (resp. -0.169), whereas the less negative (resp. positive) bias takes place in spring (resp. fall) and equals -0.033 (resp. 0.027). CO mixing ratio

is slightly overestimated in winter and spring similarly (MNMB = 0.098), with lower biases during summer (-0.011) and fall (0.024). Nevertheless, all MNMB and FGE are very low, showing good skills from the MOCAGE REF-C1SD simulation.

Table 2 compares the assessment of MOCAGE-M versus IAGOS-DM with the assessment of MOCAGE-M-noW versus IAGOS-DM-noW versions. This comparison is based on the IAGOS-DM-noW sampling level. In Table 2, the comparison between the two methods shows a better agreement between the model and the observations when we apply the interpolation

with the weighting factors. The $O_3$ correlation with the "noW" products decreased to 0.84–0.92 compared to the 0.90–0.95 derived from our method, and the CO correlation dropped from 0.72–0.81 down to 0.63–0.72. The MNMB and the FGE show better scores for the "noW" products in each case, except the $O_3$ MNMB in DJF. The general improvement of normalised biases, normalised errors and spatial correlations, compared to a simplified gridding method, suggests that the use of a weighting function in our methodology can significantly enhance the model assessment.

### 4.3   Regional-scale analysis

In this section, we attempt to evaluate the simulation in the UT and the LS separately, focusing on the seasonal cycles. For this, we sort both data sets between the two layers as explained in Sect. 3.4. As a first step, before comparing the simulation to the observations, we analyse the impact of the mapping method for IAGOS onto the MOCAGE grid on a monthly basis. For this purpose, two versions of the IAGOS data set are used. Hereafter, IAGOS-HR refers to the high-resolved IAGOS data

synthesized in Cohen et al. (2018), where every single measurement was categorized as belonging to the UT ($P_{TP}$ +15 hPa < P < $P_{TP}$ +75 hPa), the tropopause transition layer or the LS (P < $P_{TP}$ -15 hPa), and where regional monthly means were derived by averaging all the concentrations measured above the defined region. In contrast, IAGOS-DM refers to the new product presented in this paper, i.e. the IAGOS data distributed onto the model's grid, then assigned into either the UT or the LS based on the monthly averaged PV at each model grid point. Note that IAGOS-HR seasonal cycles were computed on the original

regions' coordinates, but the changes induced by the 1° difference in some of the regions are expected to be negligible, based on the geographical sensitivity tests mentioned in Cohen et al. (2018).

The comparison between the two IAGOS products in matter of seasonal cycles is proposed in Figs. 7 and 8, respectively for $O_3$ and CO. They are shown with their corresponding interannual variability (IAV), defined as a year-to-year standard

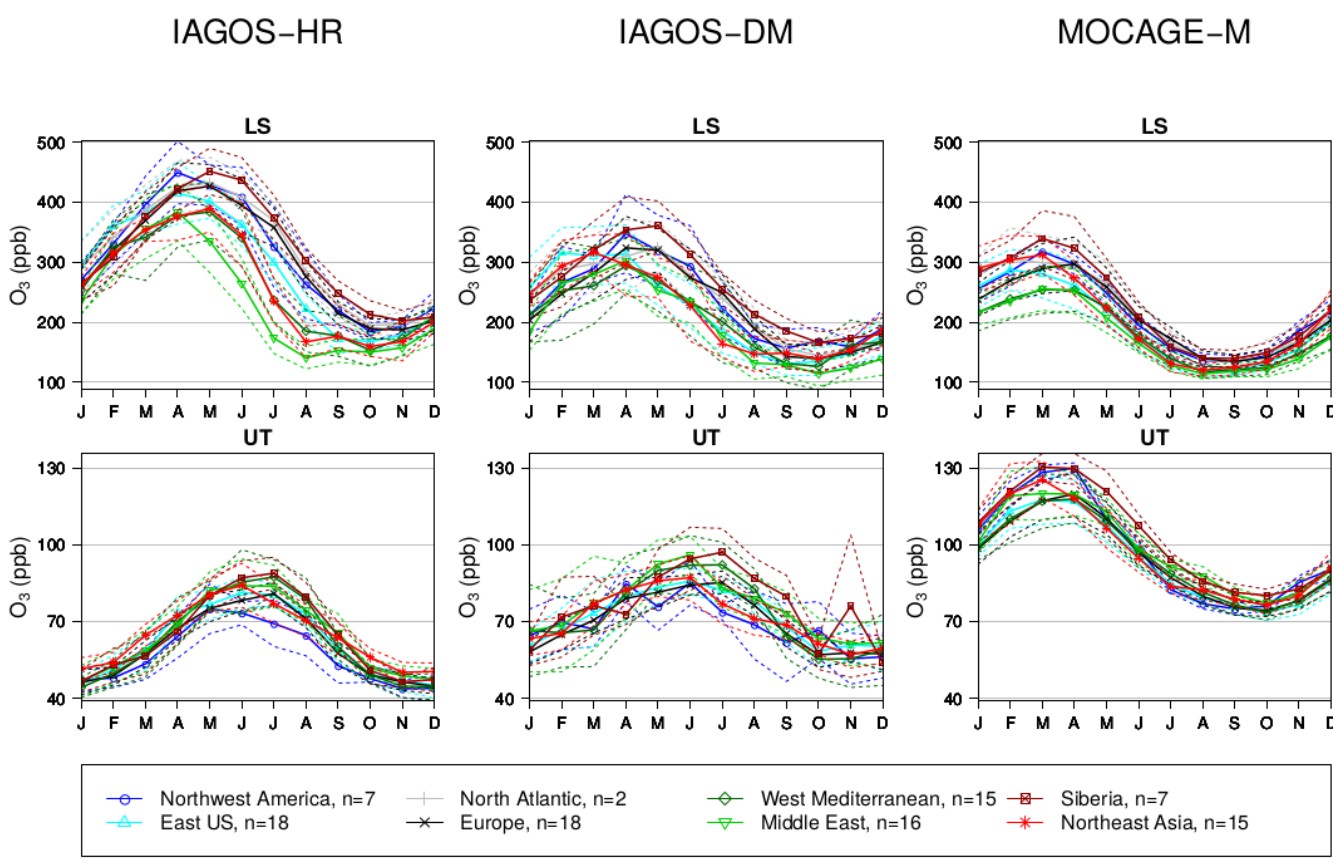

**Figure 7.** Mean seasonal cycles in ozone above the eight regions from 1995 until 2013, in the UT for the bottom panels, and in the LS for the top panels. Solid and dashed lines correspond respectively to mean values and their interannual variabilities, respectively. The left column corresponds to the high-resolution IAGOS data set (IAGOS-HR) presented in Cohen et al. (2018), and the column at the middle to IAGOS-DM. The right column represents the cycles derived from the simulation (MOCAGE-M), using the same grid points as in IAGOS-DM. The legend is shown at the bottom. For each region, the n integer indicates the amount of selected years contributing to the IAGOS-DM mean seasonal cycles in the UT.

deviation. For complementary information, a more exhaustive representation is proposed in Figs. C1 and C2 in the Appendix showing the results with each region in a distinct panel. In Fig. 7, both IAGOS versions show a summertime $O_3$ maximum in the UT and a springtime maximum in the LS. A lessened contrast between the UT and the LS is observed in IAGOS-DM. In the UT, the $O_3$ volume mixing ratio and its interannual variability are higher in IAGOS-DM than in IAGOS-HR for winter and

5    fall seasons ($\sim 60 \pm 20$ ppb compared to $\sim 50 \pm 10$ ppb), whereas they are similar in spring and summer. In this layer, the most important differences between the two versions thus take place during lower-ozone seasons. In the LS, the $O_3$ amounts are lower in IAGOS-DM ($\sim 110$–375 ppb) than in IAGOS-HR ($\sim 150$–450 ppb) during the whole year. There are two main reasons that explain the lower $O_3$ amounts in the LS and the higher amounts in the UT in IAGOS-DM compared to IAGOS-

HR. First, the projection of IAGOS observations with a very fine vertical resolution onto the MOCAGE vertical grid with a ∼800 m vertical resolution. Second, the use of a monthly PV cannot provide the description of the day-to-day variations of the tropopause altitude, whereas the latter can be important to sort the data points between the two layers. In other words, the effect of time averaging leads to a loss of tropopause sharpness, thus resulting in a mis-classification of a non-negligible part

of the individual measurements. For a given layer, it introduces a bias due to unexpected mixing with another layer. Figure 7 also makes possible to compare the behaviour of each region. In the LS, the differences between northern and southern regions shown in IAGOS-HR are generally also visible in IAGOS-DM. The regional behaviours discussed in Cohen et al. (2018), i.e. the low summertime $O_3$ mixing ratio in the Northwest American UT and in the Middle East LS, remain visible in IAGOS-DM, although the last one is substantially less pronounced. We also note high ozone values in November in the Siberian UT seen

by IAGOS-DM only. It is linked to a strong positive anomaly in November 1997 due to an upper-layer air mass that could not be differentiated to the UT, and weakly balanced by the average with too few other years.

In Fig. 8, the CO seasonal cycles in the UT are consistent between IAGOS-HR and IAGOS-DM, with a generally low differ­ence, a common springtime maximum, and a consistent inter-regional variability: a higher CO level in the two regions on the Pacific coast (Northwest America and Northeast Asia), higher summertime amounts in Northeast Asia, and lower CO levels

in one of the two southernmost regions (Middle East). Note that the monthly resolution of both PV and filtering leads to a lessened sampling in the UT in IAGOS-DM. In the North Atlantic region where aircraft trajectories describe a narrow altitude range, the resulting seasonal cycles were incomplete so that we chose to exclude them from both figures. We applied the same treatment to CO in the UT above the West Mediterranean basin and Siberia, where the level of sampling during winter and spring (not shown) are insufficient to provide complete seasonal cycles.

In the LS, the CO mixing ratio is always higher in IAGOS-DM (∼50 ppb–95 ppb) than in IAGOS-HR (from ∼40 ppb up to 65 ppb). In IAGOS-HR, a seasonal cycle is noticeable only in Middle East and Northeast Asia, whereas it is the case for almost every region in IAGOS-DM. The influence of the troposphere is increased in IAGOS-DM, with a high peak in May for the West Mediterranean basin, in June–July for Northeast Asia and in July for Siberia, likely related to the effects of boreal biomass burning in the latter. Thus, mapping the observations onto the model grid changes significantly the CO seasonal cycles

in the LS.

As for $O_3$, the reason why the CO amounts in IAGOS-DM are higher in the LS and lower in the UT comes from the coarse vertical resolution in the MOCAGE grid, and from the uncertainty when sorting the UT data from the LS data using a monthly mean modelled PV field. More generally, the comparison between IAGOS-HR and IAGOS-DM for $O_3$ and CO clearly shows that the processing applied for mapping IAGOS high resolution data set onto the MOCAGE coarse grid slightly modifies IA-

GOS characteristics. This processing, which enables a meaningful comparison between IAGOS long-term measurements and the REF-C1SD simulation, acts as a numerical filter. It is important to note that the seasonal cycles in IAGOS-DM generally show values ranging between the MOCAGE-M and the IAGOS-HR cycles, such as the yearly means in Table 3, especially in the LS where the mean $O_3$ bias drops from 84 ppb with IAGOS-HR down to 19 ppb with IAGOS-DM. The correlation in time also tends to be enhanced by the use of the IAGOS-DM product. It confirms that the representation derived from IAGOS-HR cannot be reached by a model with the typical REF-C1SD resolution, especially for CO in the LS, but some main characteristics mentioned above can still be used as criteria. Last, the comparison synthesized in Table 4 also shows a better consistency between model and observations when our method is applied, mainly in matter of biases in the LS. No significant change is observed in the UT.

We now assess the MOCAGE-M seasonal cycles by comparing them to IAGOS-DM. As complements to Fig. 7, statistical results are given in Table 3. Note that averages calculated over all represented regions have been computed only to synthesize the assessment and to provide a quantification that confirms some features seen in the figures. As they are similar to the zonal averages, they are not meant to have a geophysical signification. A qualitative summary is also provided in Table 5. In the UT, MOCAGE-M shows a springtime maximum and higher $O_3$ concentrations (from ∼120 ppb up to 130 ppb), instead of the observed summertime maximum (season which $O_3$ values range between ∼80 ppb and 110 ppb). Adding the fact that simulated $O_3$ levels are particularly strong in the northernmost regions (Western North America and Siberia) where the stratosphere at the cruise levels is richer in $O_3$, it is likely that the stratospheric influence on the UT is overestimated in the simulation. The inter-regional averages shown in Table 3 confirm the significant difference between the two data sets in the UT, both from the $O_3$ mixing ratio ($97 \pm 5$ ppb in MOCAGE-M compared to $72 \pm 9$ ppb in IAGOS-DM) and from the seasonality ($\bar{r} = 0.35$). In the LS, the simulation reproduces well the cycles including the seasonality ($\bar{r} = 0.84$ as shown in Table 3), the magnitude, the amounts of ozone ($203 \pm 23$ ppb compared to $222 \pm 36$ ppb from IAGOS-DM) and the inter-regional differences. The latter are characterized in both data sets by lower ozone levels in the two southernmost regions (West Mediterranean basin and Middle East) and higher ozone levels in the two northernmost regions (Western North America and Siberia). Without the noisy signal characterising Western North America and the West Mediterranean basin in IAGOS-DM, the interval representing the springtime interannual variabilities spreads from ∼200 ppb up to ∼400 ppb in both data sets, showing another feature well reproduced by the model. Though on a yearly basis, according to Table 3, the model tends to underestimate ozone IAV on average by a factor 1.6.

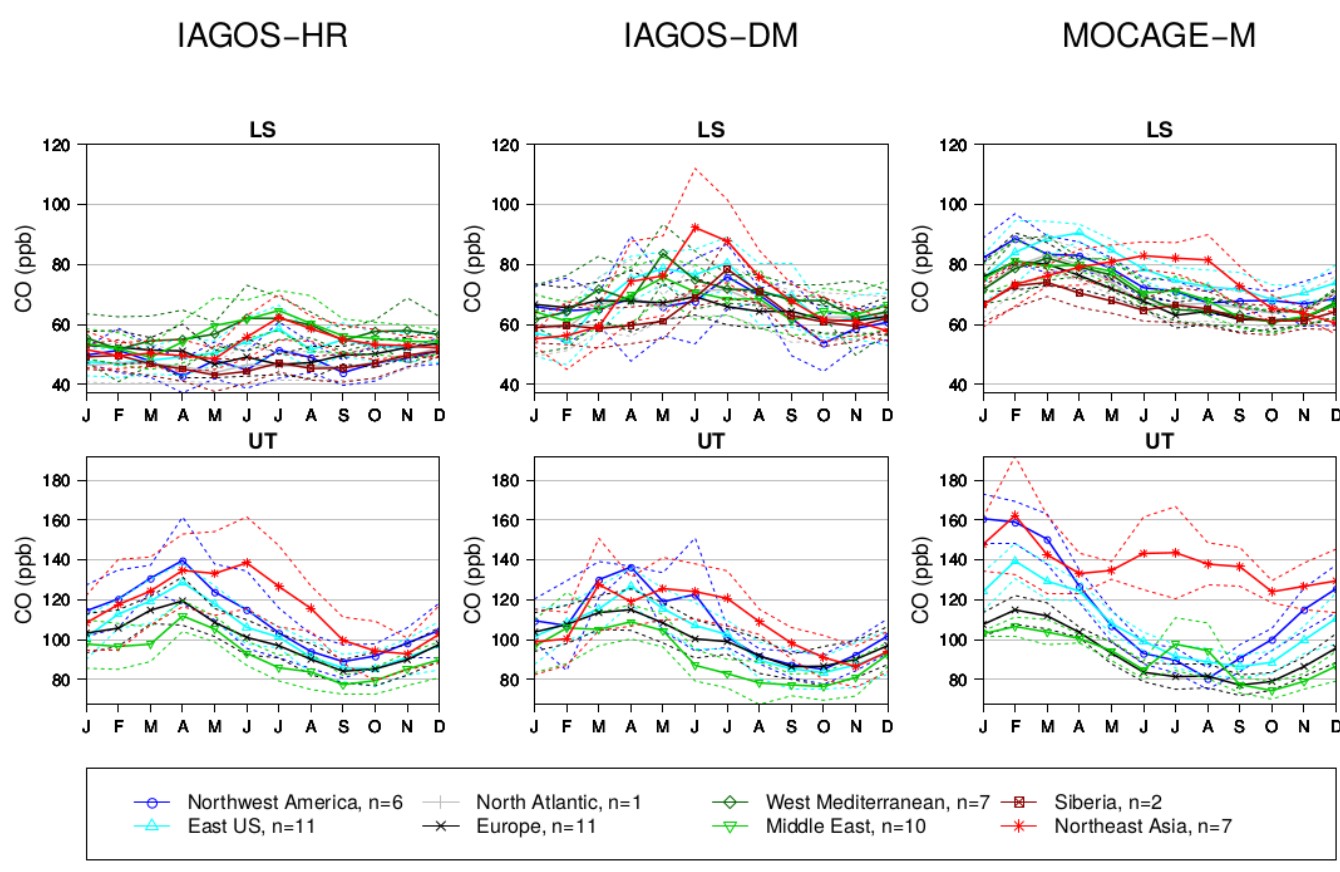

**Figure 8.** Same as Fig. 7 for carbon monoxide, from 2002 until 2013.

**Table 3.** Cross-regional averages derived from the seasonal cycles. Yearly mean values and interannual variabilities are shown for IAGOS-HR, IAGOS-DM and MOCAGE-M, along with the correlation coefficient between MOCAGE-M and the two IAGOS versions.

| Species | Layer | IAGOS-HR | IAGOS-DM | MOCAGE-M | $\bar{r}_{\text{IAGOS-HR}}$ | $\bar{r}_{\text{IAGOS-DM}}$ |
|---------|-------|----------|----------|----------|-----------------------------|-----------------------------|
| $O_3$ (ppb) | LS | $287 \pm 33$ | $222 \pm 36$ | $203 \pm 23$ | 0.70 | 0.84 |
|  | UT | $62 \pm 6$ | $72 \pm 9$ | $97 \pm 5$ | 0.14 | 0.35 |
| CO (ppb) | LS | $51 \pm 5$ | $66 \pm 8$ | $72 \pm 5$ | 0.28 | 0.31 |
|  | UT | $104 \pm 11$ | $101 \pm 11$ | $109 \pm 8$ | 0.63 | 0.58 |

**Table 4.** As in Table 3 for IAGOS-DM and MOCAGE-M derived without weighting (IAGOS-DM-noW and MOCAGE-M-noW).

| Species | Layer | IAGOS-DM-noW | MOCAGE-M-noW | $\bar{r}_{\text{noW}}$ |
|---------|-------|--------------|--------------|------------------------|
| $O_3$ (ppb) | LS | $235 \pm 39$ | $181 \pm 20$ | 0.84 |
|  | UT | $72 \pm 9$ | $98 \pm 5$ | 0.34 |
| CO (ppb) | LS | $63 \pm 8$ | $77 \pm 5$ | 0.25 |
|  | UT | $101 \pm 11$ | $110 \pm 9$ | 0.54 |

**Table 5.** Synthesis of MOCAGE-M ability to reproduce the main features from IAGOS-DM.

| Species | Layer | Main features from IAGOS-DM | Reproduced by MOCAGE-M |
|---------|-------|------------------------------|------------------------|
| $O_3$ | LS | Springtime maxima | Yes |
| | | Northward gradient | Yes |
| | UT | Summertime maxima | No |
| | | Less $O_3$ in WNAm summer | No |
| | | More $O_3$ in Sib | Yes |
| CO | LS | Summertime maxima | No |
| | | More CO in WMed | No |
| | | More CO in NEAs summer | Yes |
| | UT | Springtime maxima | Yes (winter-spring) |
| | | More CO in WNAm and NEAs | Yes |
| | | Spring-summer maximum in NEAs | No |

The modelled CO seasonal cycles (Fig. 8) in the UT show similarities with the observations (IAGOS-DM), including the higher concentrations in the two Pacific coast regions (Western North America and Northeast Asia), the strong summertime concentrations in Northeast Asia and also comparable mixing ratios between the model and the IAGOS-DM observations in most regions, as confirmed by Table 3. However, the simulation overestimates the CO mixing ratios in the two Pacific

coast regions, and the seasonal maxima generally take place during late winter–early spring in the simulation, earlier than the observed middle-of-spring maxima. The seasonal minima are in phase with the observations. In the LS, the seasonal cycles magnitude is underestimated by the simulation but the overall bias remains relatively low, with a $73 \pm 5$ ppb average for MOCAGE-M compared to $69 \pm 9$ ppb for IAGOS-DM. In most regions, MOCAGE-M shows seasonal cycles in the LS in phase with the UT, thus contrasting with the observations and making the correlation drop from 0.64 in the UT to 0.31 in

the LS. This suggests that the model simulation is affected in the LS by transport from the troposphere during springtime. Consistently with observations, MOCAGE-M shows a summertime maximum in Northeast Asia exclusively. Although part of this feature may originate from the positive bias in the UT, the fact that it only concerns the summer season, in contrast to the UT, suggests that summertime convection also plays a non-negligible role.

## 5 Summary and conclusions

We developed a methodology that makes the IAGOS database ready to assess chemistry-climate long-term model simulations for recent decades, and particularly the REF-C1SD experiment produced in the frame of the CCMI phase-I project. The current paper describes this methodology and its application on a chosen simulation (the REF-C1SD simulation from MOCAGE-

CTM), assessing modelled ozone and carbon monoxide monthly fields during Aug. 1994–Dec. 2013 and Dec. 2001–Dec. 2013, respectively.

The first step consists of generating a gridded monthly IAGOS data set (IAGOS-DM), firstly by a linear-distanced reverse interpolation on the chosen model grid on a monthly basis, and then by deriving weighted monthly means on each grid cell. The second step consists of deriving seasonal and annual climatologies for the well-sampled vertical grid levels, then to derive statistical scores for the simulation assessment. In the case of the REF-C1SD simulation from MOCAGE, the yearly mean spatial distribution is well reproduced by the model, especially for $O_3$, and especially at the sampled highest levels too. It suggests a particularly good representation of the main stratospheric processes that affect $O_3$ in the UTLS. The extreme mean CO mixing ratios observed above the strongly emitting and convective regions in the tropics and subtropics are also visible in the simulation, with a very low bias above tropical Africa and a significant positive bias above South and East Asia. Globally, the annual $O_3$ normalized mean bias is very low (MNMB = -0.012) and slightly higher seasonally. They are a bit higher in winter and summer (| MNMB | = 0.144–0.169) than in spring and fall (| MNMB | = 0.027–0.033), with quasi opposite values in each couple of antagonist seasons. The yearly bias in CO is positive (MNMB = 0.049), with highest values similarly in winter and spring, and particularly low values in summer and fall. The statistical metrics were applied for each vertical grid level separately in order to locate strengths and weaknesses of the model, but also for the whole UTLS grid cells for the purpose of a bulk comparison that could be reiterated on other model simulations.

Another step consists of a comparison of the seasonal cycles between IAGOS observations and the MOCAGE simulation in the upper troposphere (UT) and the lower stratosphere (LS). It relies on the use of a monthly mean calculated PV field to define a UT and a LS separated by a transition layer, following the same principle as in Thouret et al. (2006). The mean seasonal cycles have been compared over the eight well-sampled regions defined and analysed in Cohen et al. (2018). The application to the assessment of this REF-C1SD experiment by MOCAGE is preceded by an analysis of the changes induced in IAGOS seasonal cycles by the projection on the model monthly grid. As expected, going from IAGOS-HR to IAGOS-DM systematically leads to an increase (resp. decrease) in upper-tropospheric (resp. lower-stratospheric) $O_3$, to an increase in lower-stratospheric CO and generally to a slight decrease in upper-tropospheric CO. The use of a monthly mean PV field and the $\sim$ 800 m vertical resolution in the UTLS of MOCAGE onto which IAGOS observations are projected automatically result in an artificial increase of stratosphere–troposphere exchange. It is explained by the fact that the grid cells in the vicinity of the tropopause are crossed by both tropospheric and stratospheric air masses in the course of a month. It results in a decreased vertical gradient between UT and LS. Nevertheless, the seasonal maxima and minima become less clear but remain visible in IAGOS-DM with respect to IAGOS-HR. The hierarchy between the regions is generally conserved from IAGOS-HR to IAGOS-DM, for both chemical

species and both layers: in each of these cases, we find the same regions showing lowest/highest values between the two IAGOS representations. Also, some specific local behaviours mentioned in Cohen et al. (2018) remain visible in IAGOS-DM. Concerning $O_3$, we highlighted the consistency of the lowest quantities in the UT above Western North America and, substantially less significant, in the LS above Middle East. Concerning CO, we showed the conservation of the spring-summer maximum in Northeast Asia in the UT and its summertime maximum in the LS.

The evaluation of the MOCAGE REF-C1SD simulation (MOCAGE-M) with IAGOS-DM shows a good representation of $O_3$ in the LS in matter of seasonal cycle magnitudes and geographical variability, thus highlighting the well-reproduced main stratospheric processes. In the UT, for all the regions, the model overestimates the $O_3$ mixing ratios and shows a typical lower-stratospheric seasonality, suggesting an overestimation in the transport from the stratosphere. The modelled CO field shows similarities with the observations in the UT, with a one-month shift in the seasonal maxima. One possible reason is the decadal linear interpolation in anthropogenic emissions implemented in REF-C1SD, leading to a lack of year-to-year variability in modelled CO fields. In the LS, CO is generally higher in the simulation and shows a seasonal cycle in phase with the UT, in contrast to IAGOS-DM. It suggests an overestimated tropospheric influence in this layer during springtime.

The methodology shown in this paper has proven useful for assessing the REF-C1SD experiment from MOCAGE in the UTLS, further highlighting the model strengths and weaknesses when compared to the densest *in situ* IAGOS data set in the UTLS. Particularly, the use of the IAGOS-DM product instead of IAGOS-HR systematically reduced the biases characterizing the simulation, thus avoiding an underestimation of the model abilities to reproduce the chemical composition of the UT and the LS in a recent climate time period.

The present methodology could be easily applied to CCMI REF-C1SD simulations from other models, both for an inter-model comparison and for assessing CCMI products against IAGOS database, notably intermodel-averaged fields. To a greater extent, it can be used on a wide range of long-term simulations including both CCMs nudged and free runs in order to perform climatological comparisons. Precaution must be taken while extending this work to the specified-dynamics simulations from CCMs, regarding the loss of consistency between chemical and dynamical variables that is introduced by nudging, as highlighted in Orbe et al. (2020). Notably, inconsistencies between ozone and potential vorticity are likely to introduce noise in the simulated upper-tropospheric and the lower-stratospheric behaviours. Last, the assessment illustrated in this study is based on two chosen applications of our methodology, i.e. the analyses of long-term seasonal and yearly averages on different vertical grid levels and the mean seasonal cycles in the UT and the LS, but a wide diversity of complementary comparisons remain possible. We thus recommend this new product to the CCMI community.

# Appendix A: Horizontal climatologies

## A1 Ozone

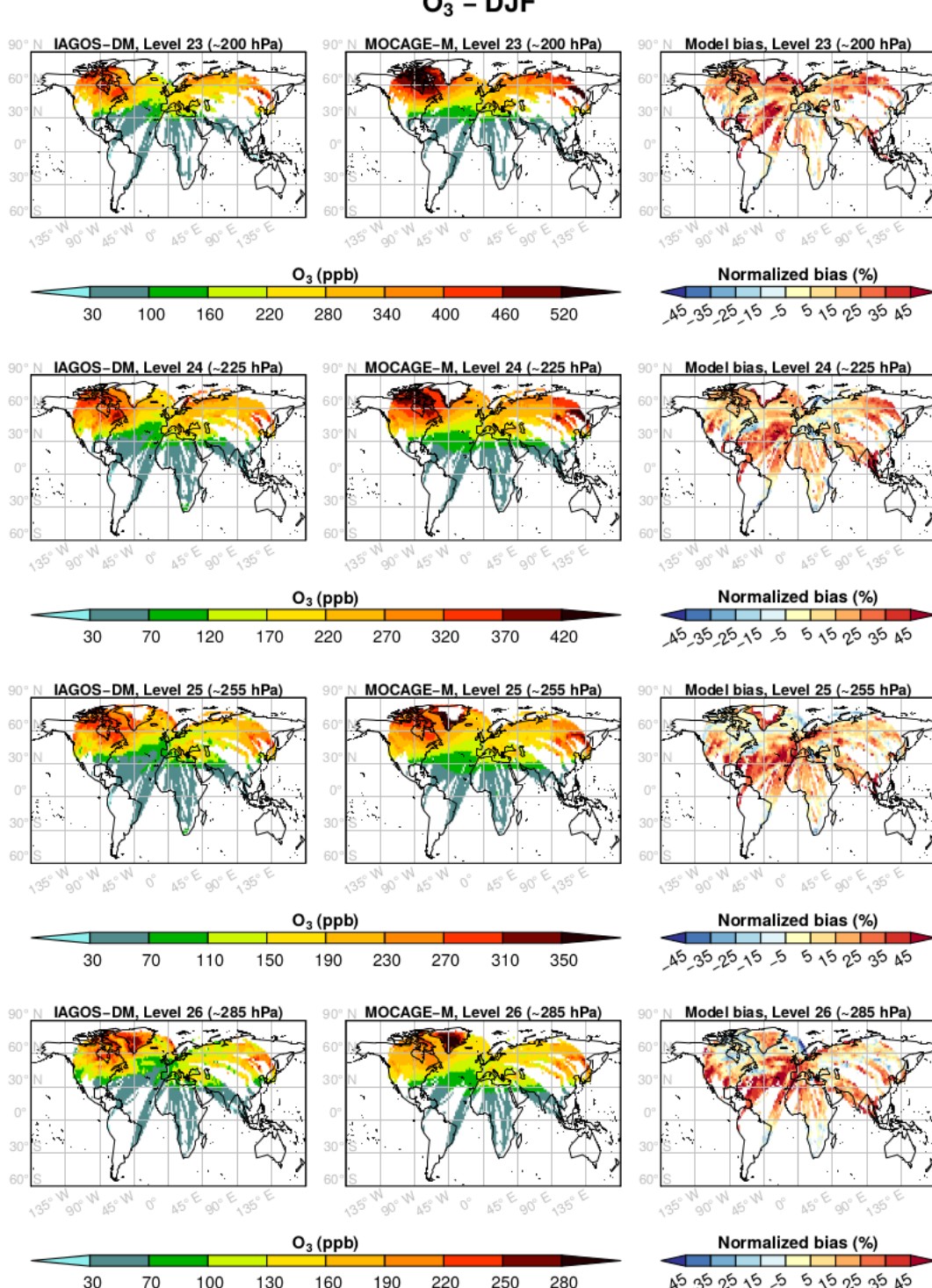

**Figure A1.** As Fig. 3 for boreal winter.

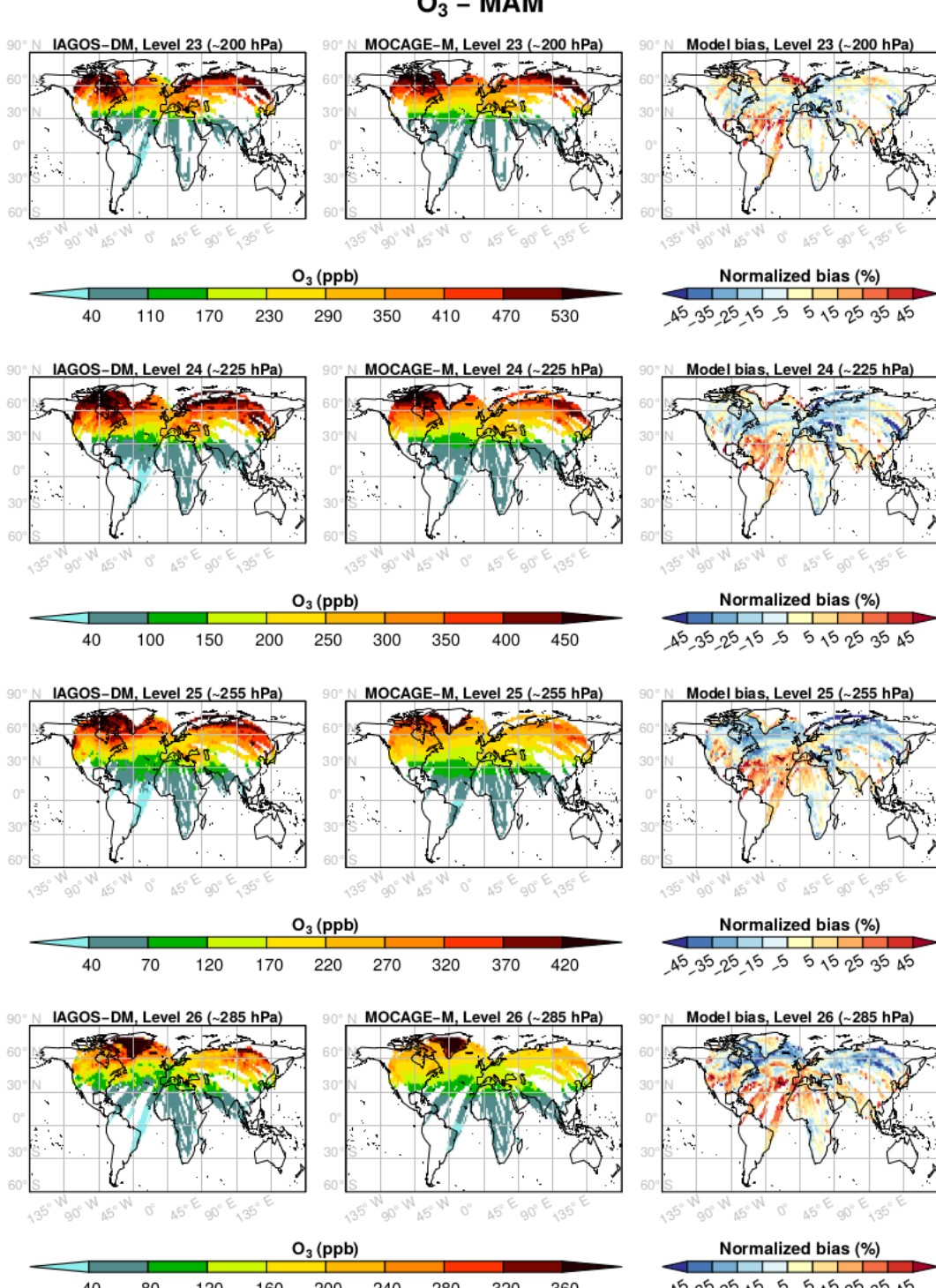

**Figure A2.** As Fig. 3 for boreal spring.

# O₃ – JJA

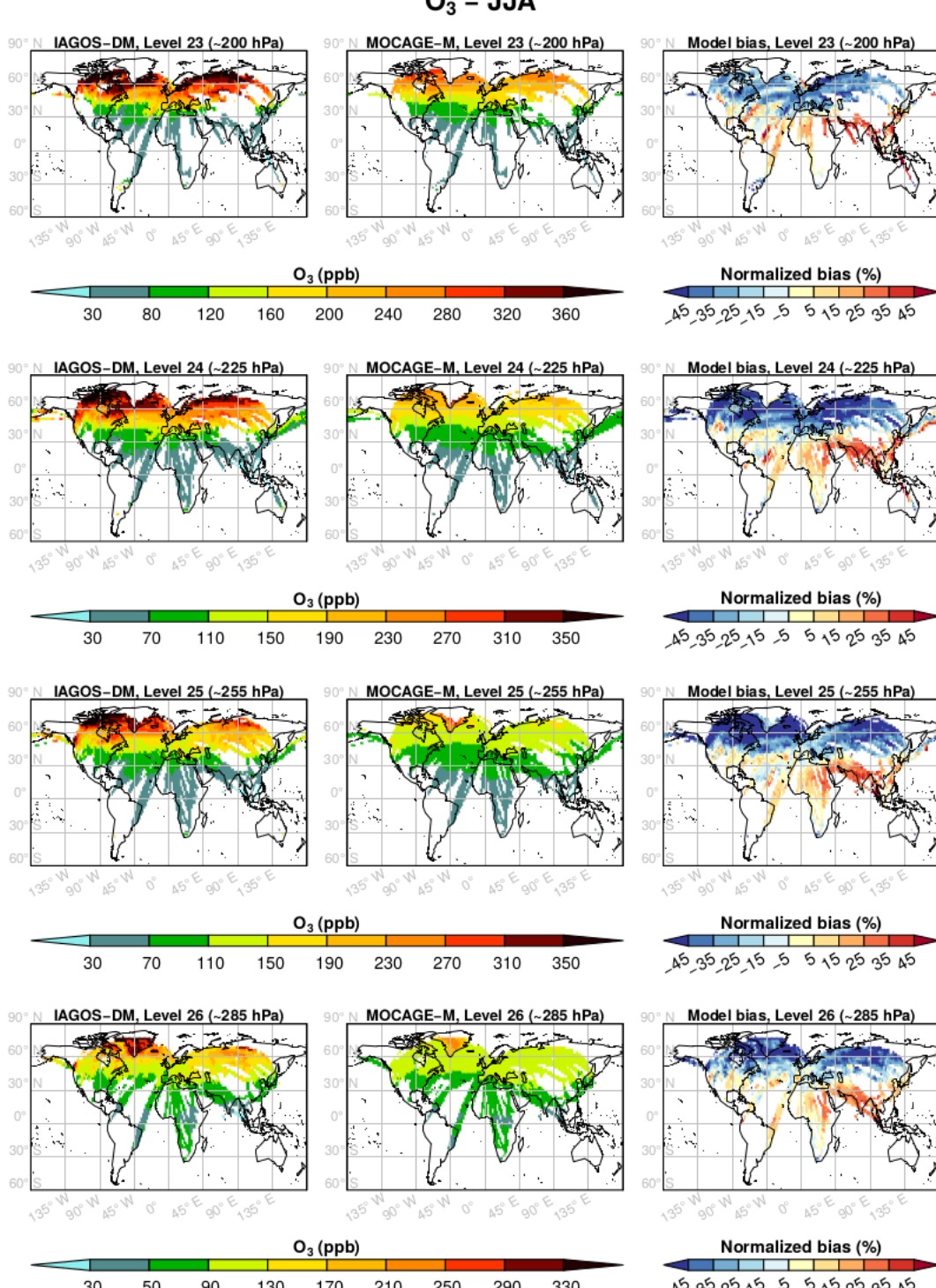

**Figure A3.** As Fig. 3 for boreal summer.

# O₃ – SON

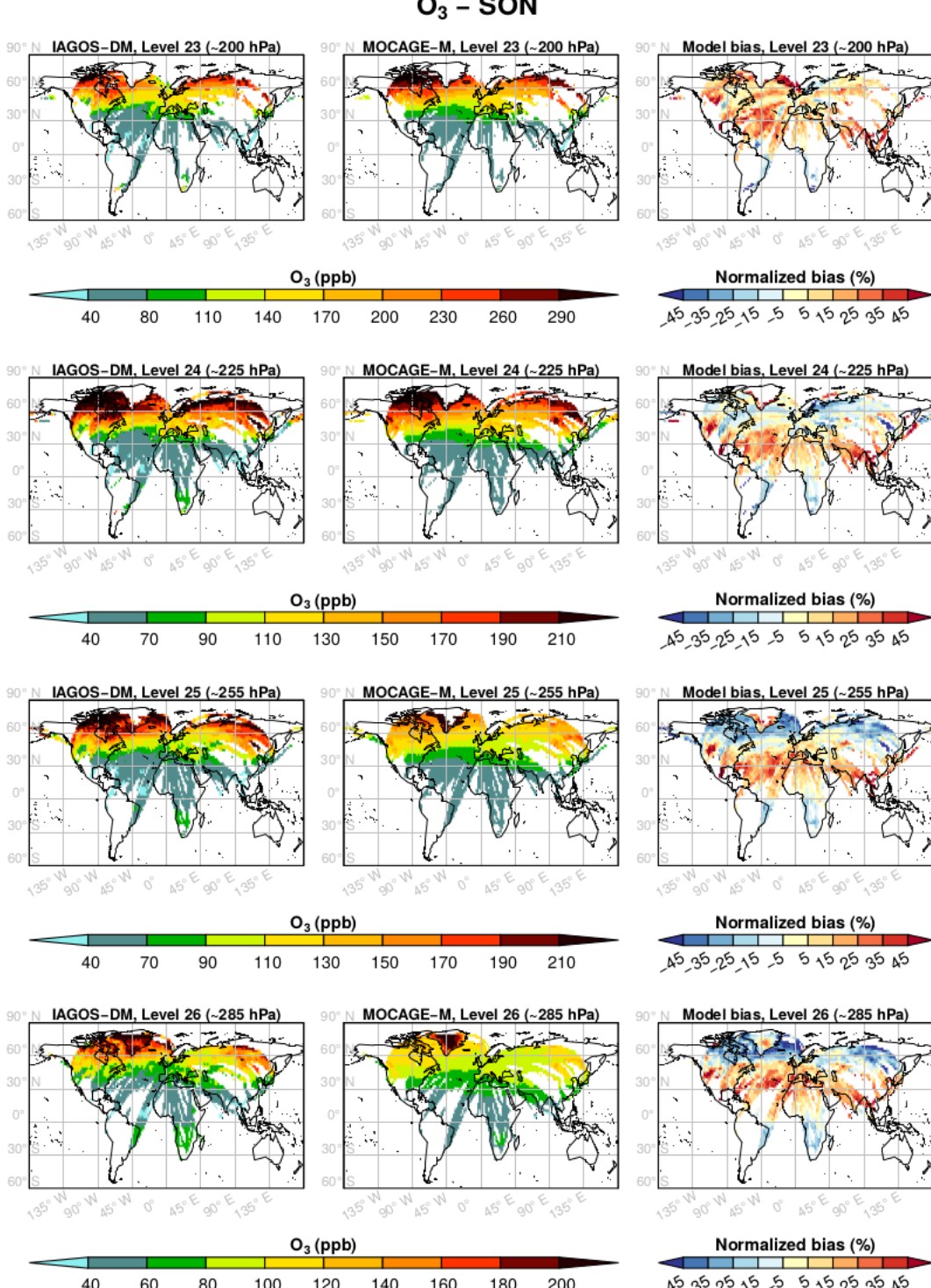

**Figure A4.** As Fig. 3 for boreal fall.

## A2 Carbon monoxide

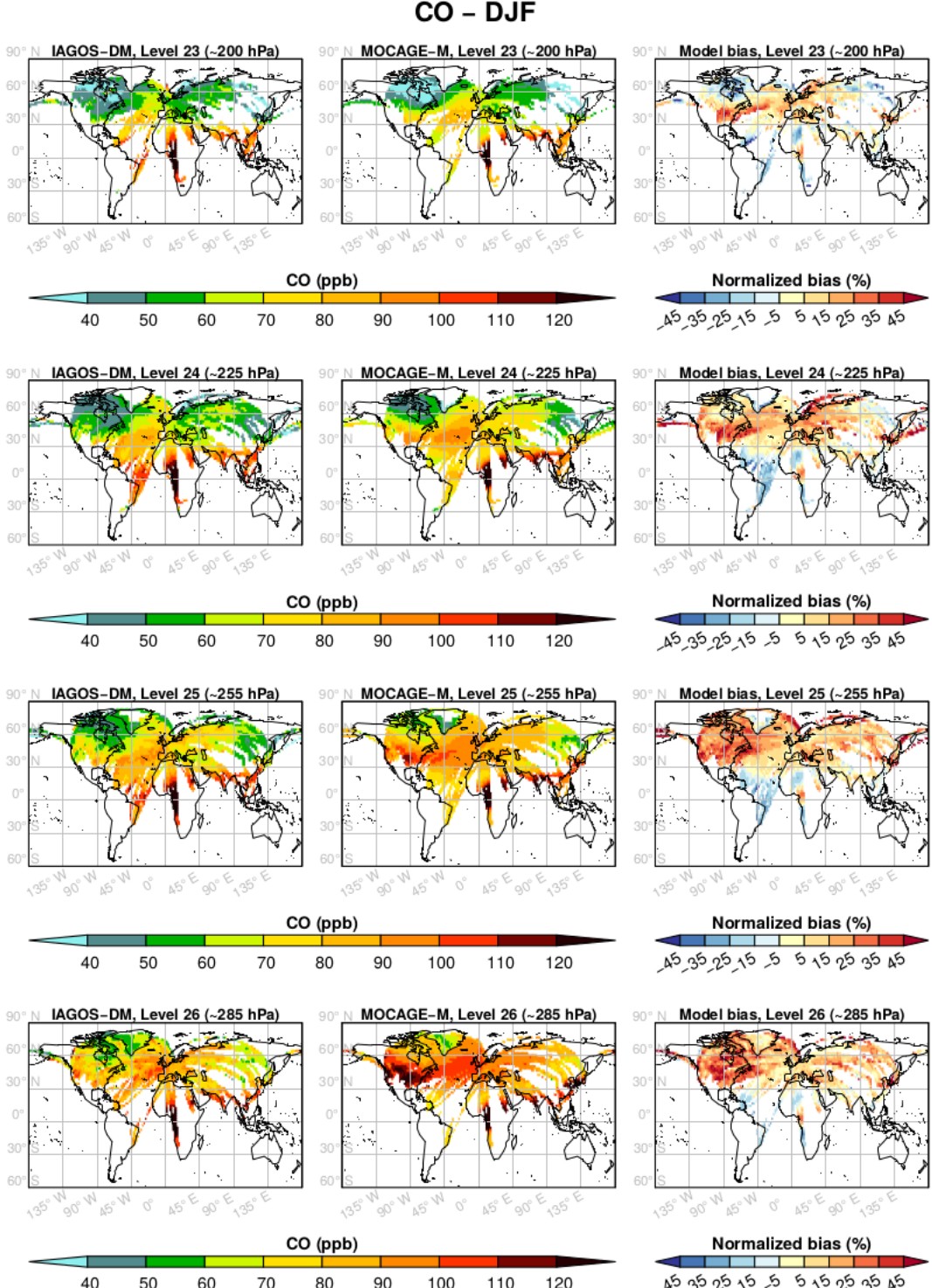

**Figure A5.** As Fig. 4 for boreal winter.

# CO – MAM

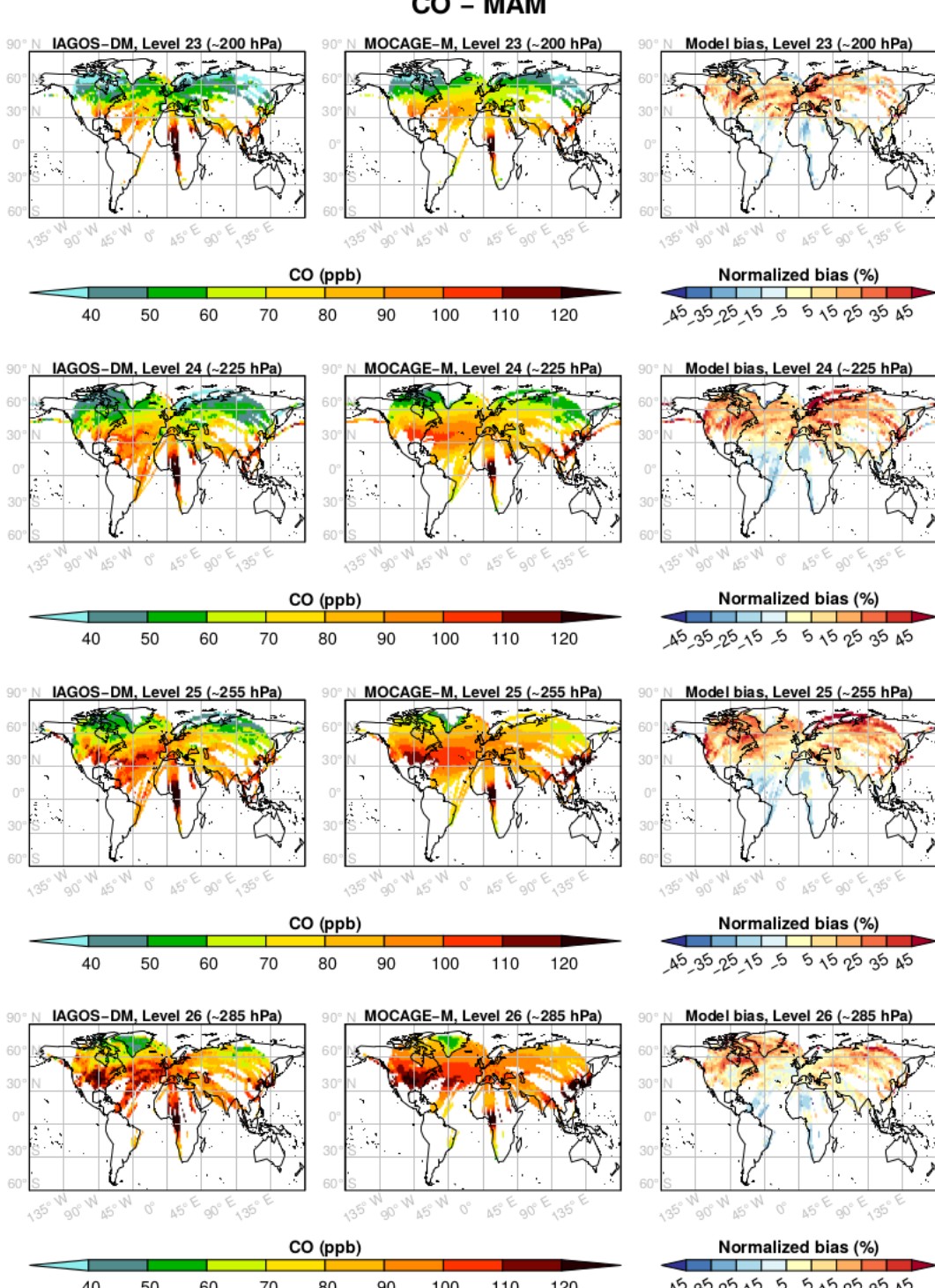

**Figure A6.** As Fig. 4 for boreal spring.

# CO – JJA

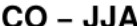

**Figure A7.** As Fig. 4 for boreal summer.

**Appendix B: Scatterplots**

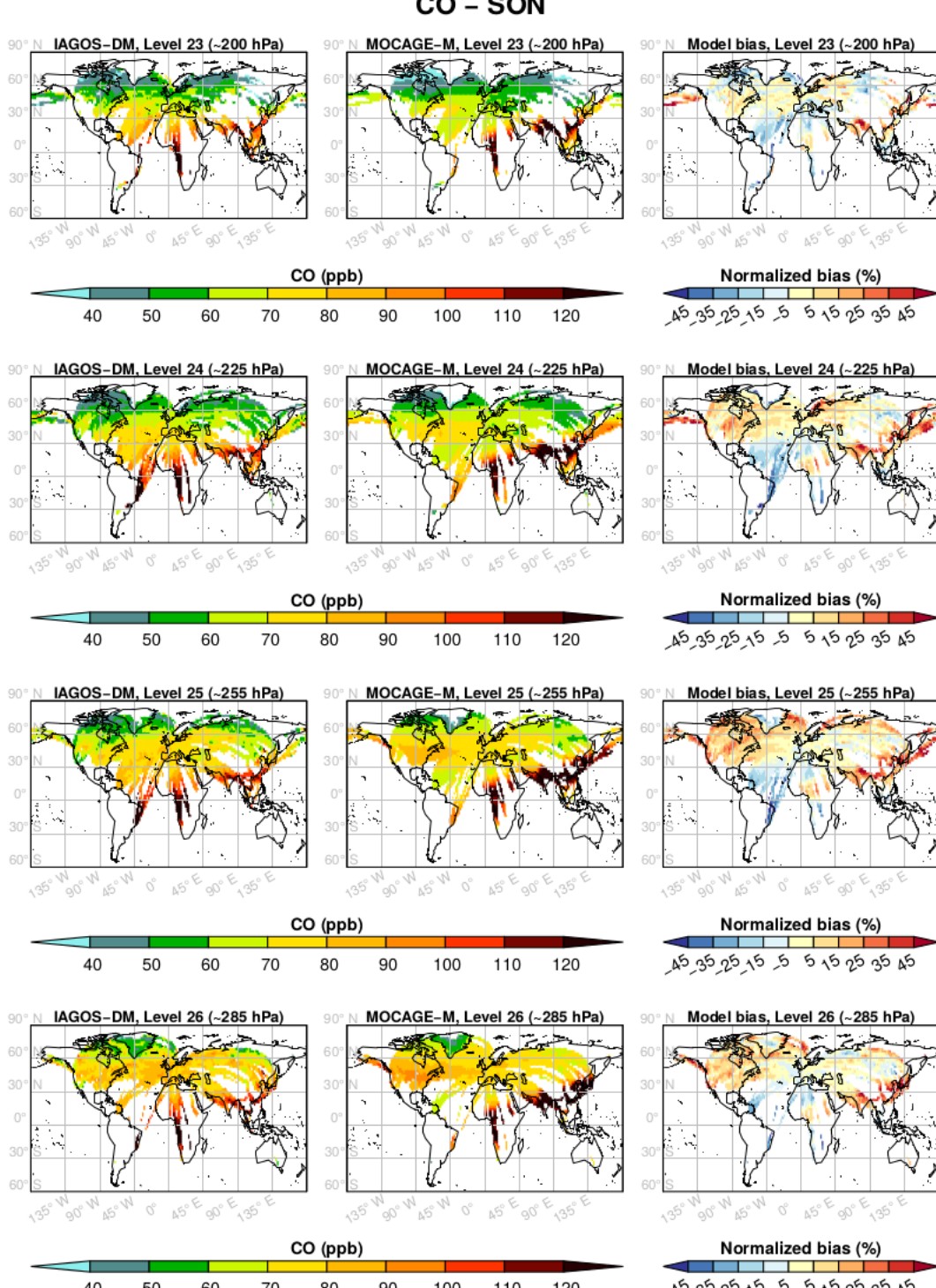

**Figure A8.** As Fig. 4 for boreal fall.

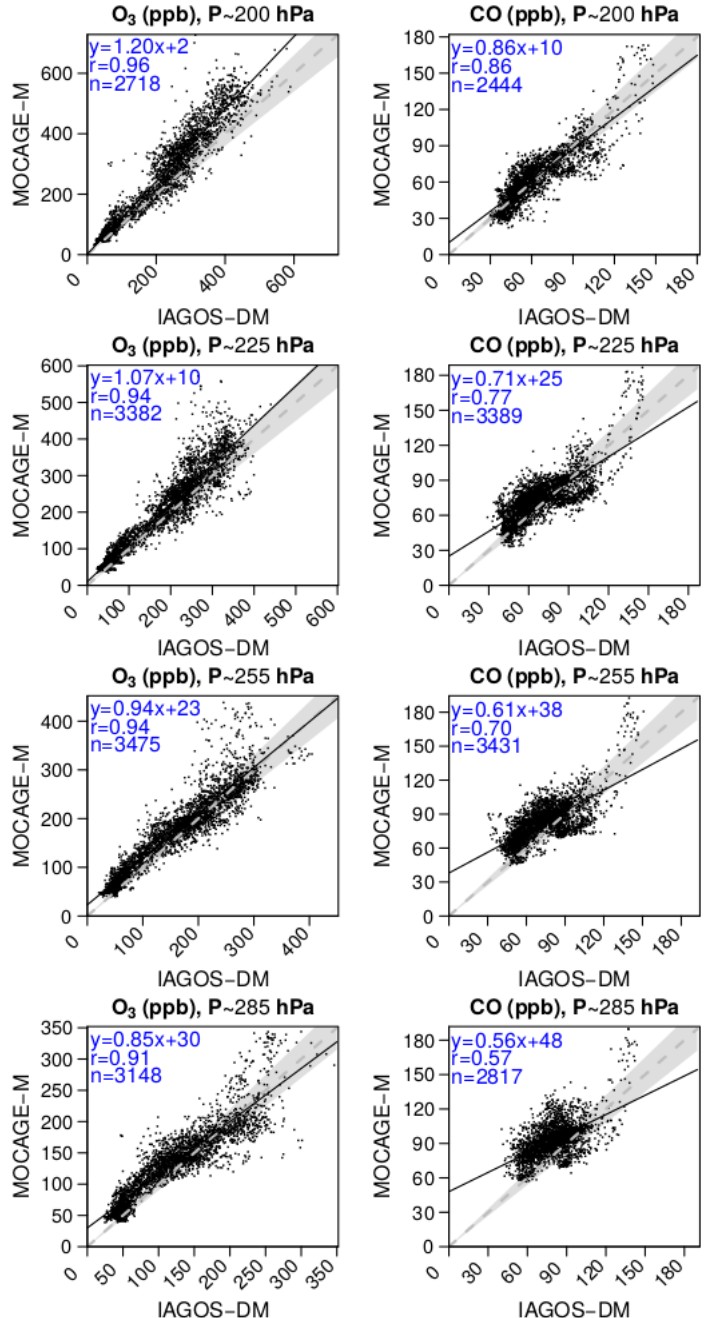

**Figure B1.** As Fig. 5 for boreal winter.

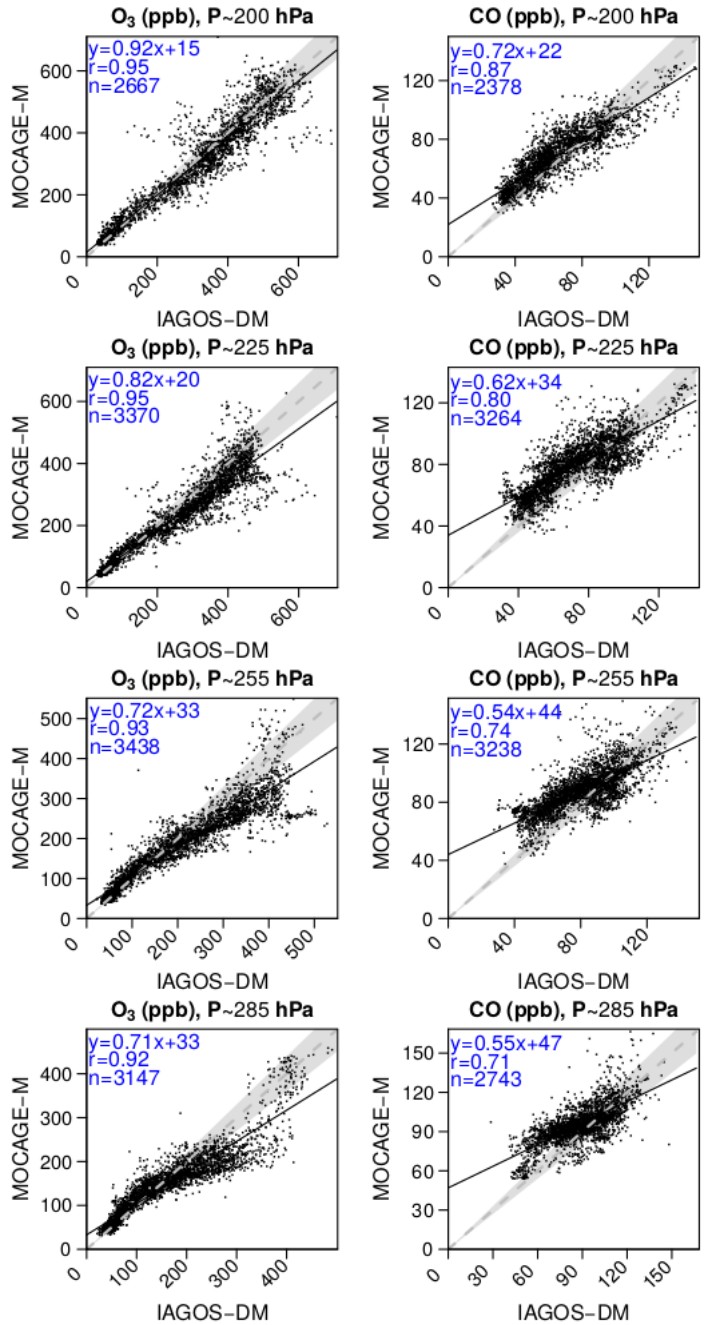

**Figure B2.** As Fig. 5 for boreal spring.

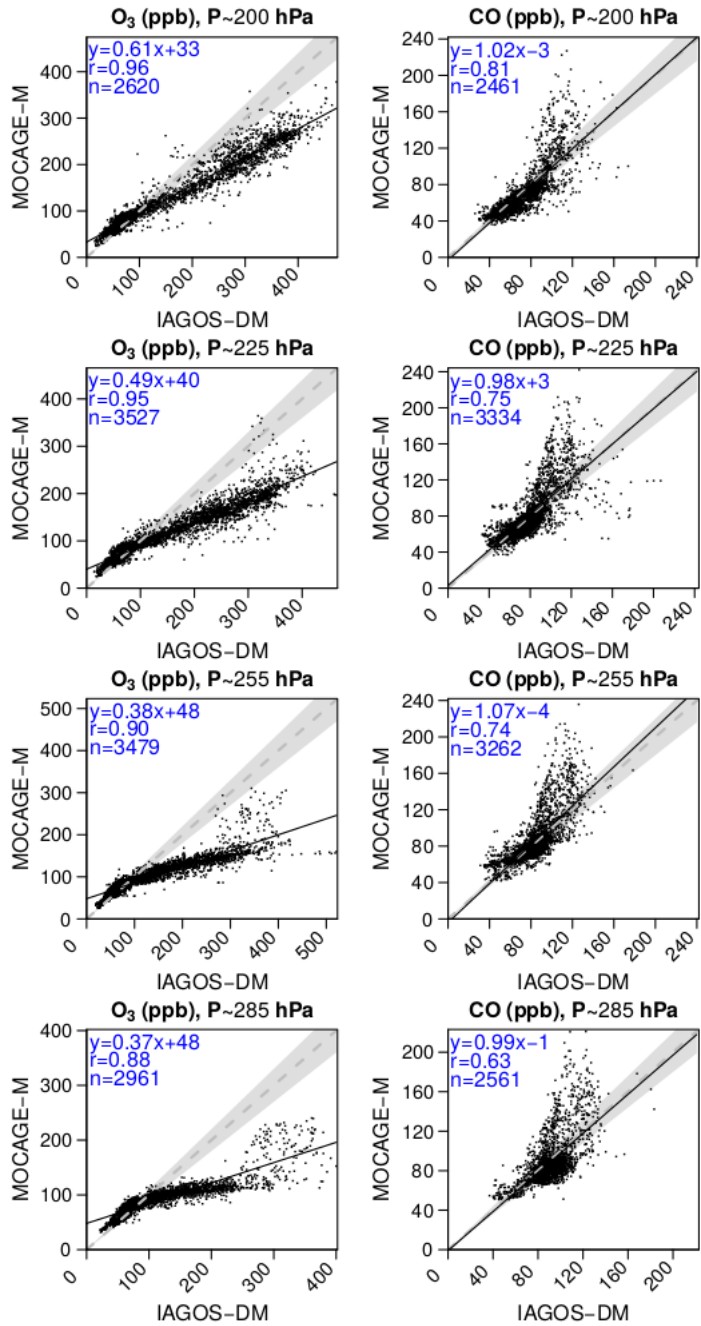

**Figure B3.** As Fig. 5 for boreal summer. The two extremely high CO values seen by IAGOS-DM above Alaska at level 25 (350 and 510 ppb) in Fig. A7 have been excluded from the scatterplots because considered as outliers.

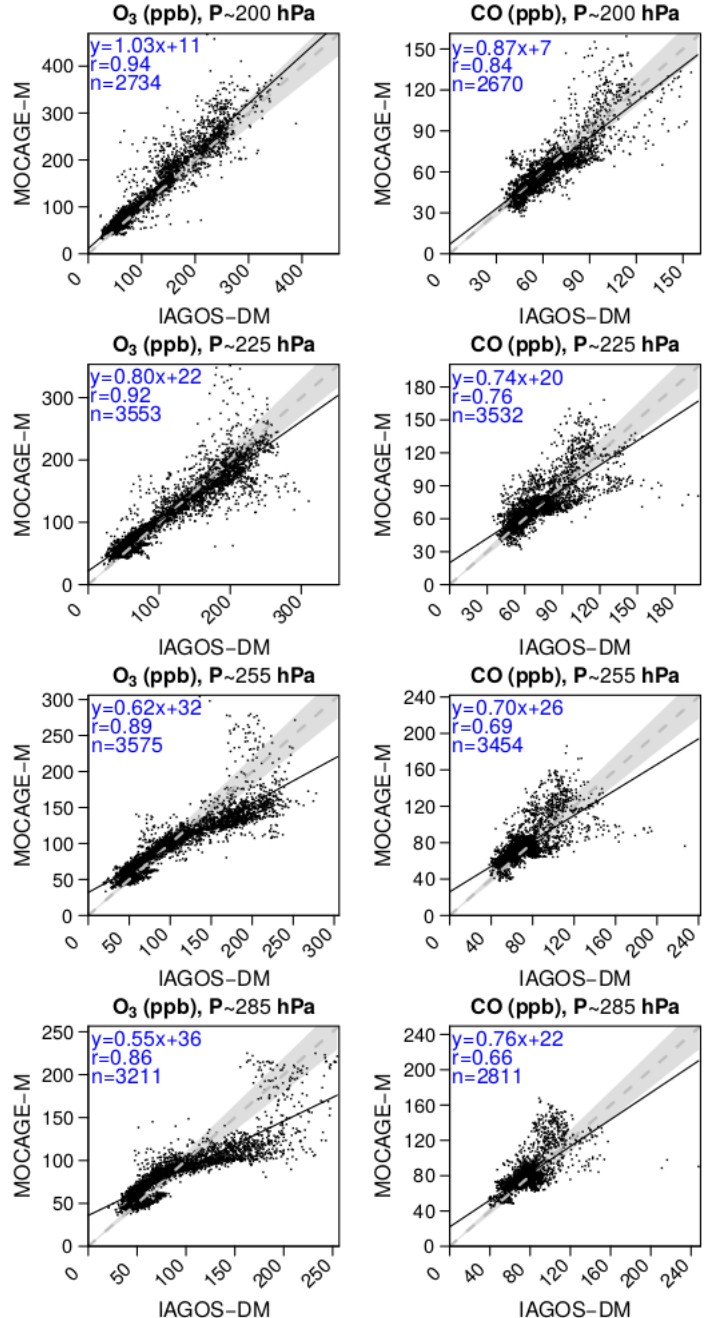

**Figure B4.** As Fig. 5 for boreal fall.

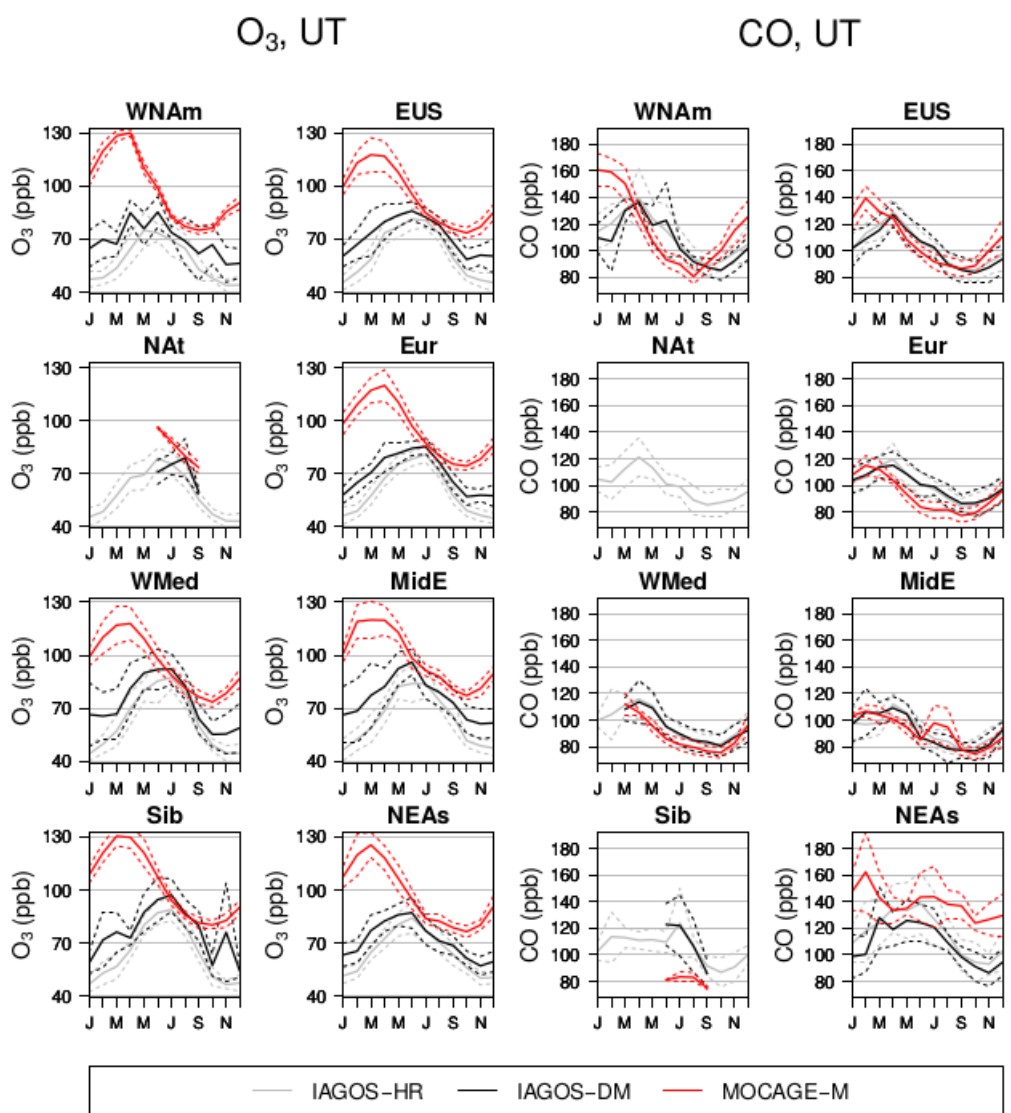

**Figure C1.** Mean seasonal cycles in the UT. $O_3$ is represented in the left two columns, and CO in the right columns. Each graphic gathers the data from IAGOS-HR (gray), IAGOS-DM (black) and MOCAGE-M (red), in one given region. The solid lines illustrate the mean values, and the dashed lines show the uncertainties regarding the interannual variability. The mean values are shown only if they are derived from several monthly means.

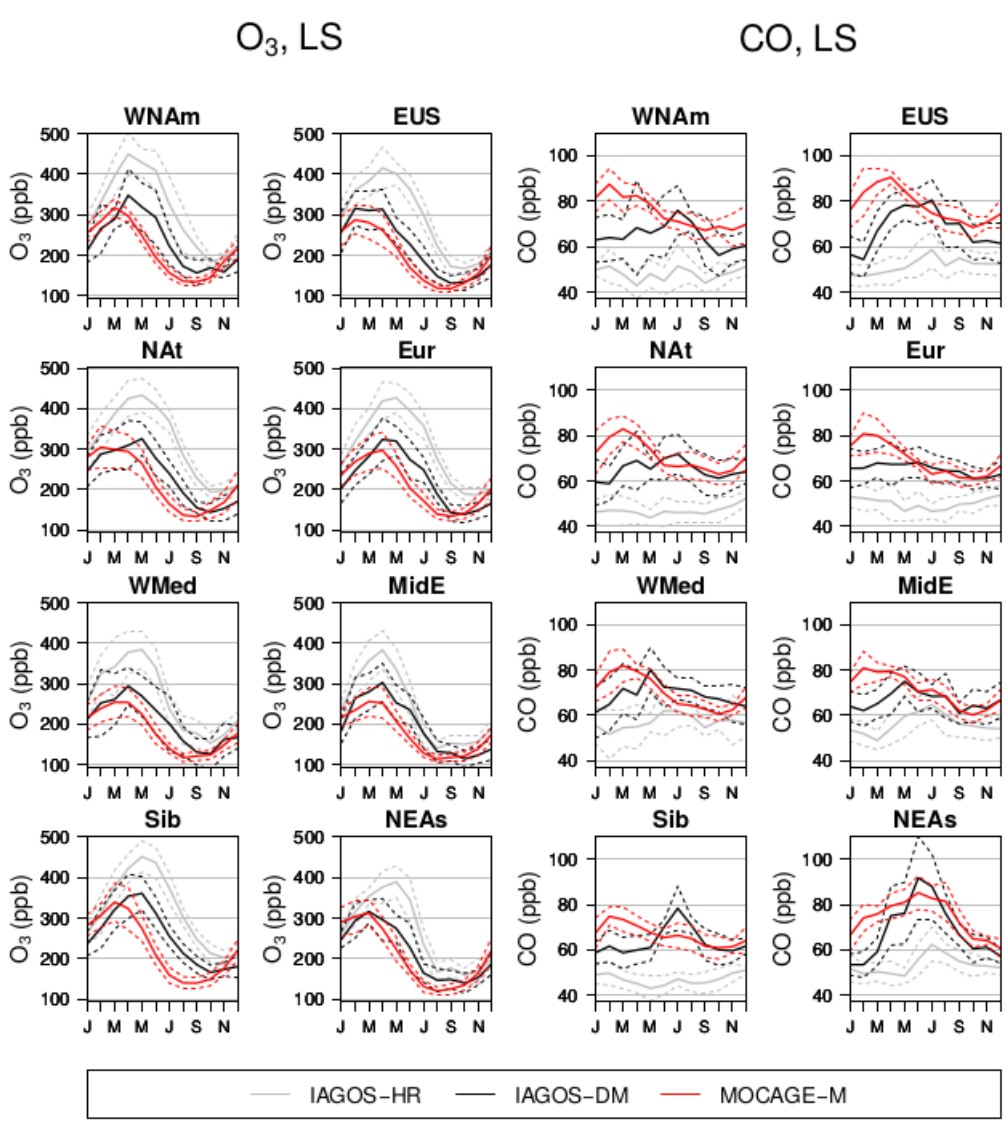

**Figure C2.** As Fig. C1 for the LS.

## Appendix C: Seasonal cycles

*Code and data availability.* The IAGOS ozone and carbon monoxide data are available at https://doi.org/10.25326/20 and more precisely, the time series data are found at https://doi.org/10.25326/06. The IAGOS-DM product adapted to the MOCAGE model grid (named IAGOS-DM-MOCAGE) is available at https://doi.org/10.25326/80 and the software (named Interpol-IAGOS) at https://doi.org/10.25326/81.

*Author contributions.* YC, VM, BJ and VT designed the study. The Interpol-IAGOS software was developed by YC. The simulation output was provided by BJ and VM, and the IAGOS data by VT. The paper was written by YC and reviewed by VM and VT, and commented, edited and approved by all the authors.

*Acknowledgements.* The authors acknowledge the strong support of the European Commission, Airbus, and the Airlines (Lufthansa, Air-France, Austrian, Air Namibia, Cathay Pacific, Iberia and China Airlines so far) who carry the MOZAIC or IAGOS equipment and perform

the maintenance since 1994. In its last 10 years of operation, MOZAIC has been funded by INSU-CNRS (France), Météo-France, Université Paul Sabatier (Toulouse, France) and Research Center Jülich (FZJ, Jülich, Germany). IAGOS has been additionally funded by the EU projects IAGOS-DS and IAGOS-ERI. We also wish to acknowledge our colleagues from the IAGOS team in FZJ, Jülich for useful discussions. The MOZAIC–IAGOS database is supported by AERIS (CNES and INSU-CNRS). Data are also available via AERIS web site www.aeris-data.fr. Yann Cohen acknowledges the University of Toulouse for providing administrative support for his PhD. The authors acknowledge the

reviewers for their fruitful comments.

*Financial support.* This research has been supported by the Occitanie region and Météo-France.

*Competing interests.* The authors declare that they have no conflict of interest.

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
