# Peer review of "Interpol-IAGOS: a new method for assessing long-term chemistry-climate simulations in the UTLS based on IAGOS data, and its application to the MOCAGE CCMI-REFC1SD simulation"

_Geoscientific Model Development, 2020_

## Referee Comment (RC1) · Anonymous Referee #1 · 9 Dec 2020

This work introduces a novel methodology and database based on the IAGOS passenger aircraft observations to evaluate chemistry-climate models in the UTLS region, where there is generally a lack of accurate observations for this purpose. This is a very valuable contribution to the effort of multi-model comparisons and the authors should consider adding their tool to the ESMVal toolbox. However, before I can recommend publication of the manuscript, I would ask the authors for some clarification of their methodology and also to strengthen their case that their rather complex approach is yielding more robust insight into model deficiencies than from more traditional approaches as outlined in the following.

Major comments

P1L12-14 The sentence 'as a first application. . .' seems confusing at this place. The authors should try to clarify (or attempt a better logical flow of) their description of the evaluation methodology they introduce. First you interpolate IAGOS onto the model grid, then you mask the model at the available IAGOS grid points, then you attempt the evaluation of the model only, right?

The authors have done a commendable job in summarising references in the introduction that use satellite observations for model comparisons using SD simulations and IAGOS particularly. However, there are earlier studies that are also highly relevant to the here proposed evaluation methodology and hence should be mentioned as well. Brunner et al. (2003, 2005), Tilmes et al. (2010), and Hegglin et al. (2010) all use aircraft (and satellite) observations to evaluate CTMs and free-running CCMs in the UTLS, respectively. At least a qualitative discussion of how your findings improve (or whether they yield similar results) compared to the latter two studies in particular is needed, given they use tropopause-based coordinates to evaluate CO and O3 in the upper troposphere and lower stratosphere to evaluate CCMs, an equivalent methodology as proposed here but applied to free-running simulations. While you may argue that your method has the advantage to have near-coincident measurements for the model evaluation, the problem with the SD-simulations could be that certain nudging methodologies may lead to introducing noise (and hence too much mixing) especially visible around the tropopause (see discussion in Orbe et al. 2020). This issue indeed could help your interpretation of your findings (too much/too little ozone in the UT and LS, respectively, and vice-versa for CO).

P4L3 ff. The statement that it is not possible is an exaggeration. As shown by Brunner et al. (2003, 2005) interpolation onto the measurement space can yield in-depth insight into model behaviour and even though the data may not be available from the

multi-model intercomparison data archives, higher resolution that would be needed for this task could be obtained from the modelling centers. In fact, it would be important to prove for a methodology paper as you have presented here that your claim of the gridded IAGOS data being representative of the monthly mean is true. You could do this by sub-sampling your model according to IAGOS measurement locations (including weighting and interpolation on measurement location), then compare the mean of this sub-sampled model field to the IAGOS-based masked model field. Agreement between the two will tell you the accuracy of the assumption underlying your methodology. This extra step in my eyes would be needed to make this methodology paper a sound contribution to model evaluation efforts, and without it the question remains whether differences in the model-measurement comparison arise from comparing apples to oranges as opposed to true model deficiencies.

On a similar note, it would also be good to see what the benefit of the weighted gridding versus a gridding of the observations without the weighting function would be (equivalent to what you have done in Section 4.2 for IAGOS-DM and IAGOS-HR). This is important in order to convince the reader that the weighted gridding process is in fact worth the effort. I would expect that the interpolation/weighting is less needed for the maps on different pressure levels, unlike the result of the tropopause-based evaluation in Section 4.2, but I could be wrong.

P9L22 I am not clear why/how the shorter period for CO measurements compared to O3 matter here? After all, 10 or 20 years may result in not so different climatologies, in fact a 10-year climatology may be more representative than a 20-year one (as for ozone) since you will find quite some trends in the latter. In any case, the determination of Nthres seems rather arbitrary. How did you determine it? Did you make a quantitative analysis by how much the mean would change if more (or less for that matter) than 100 samples were included in the mean?

Minor comments

P10L21 to P11L5-15 Please clarify/write more succinctly here that you use monthly PV fields to distinguish between UT and LS observations. Also, it is not clear to me how the additional screening is applied to the data to get rid of too high (low) values in the UT (LS) respectively (is it applied to the monthly means only or to the input data that are used to generate these monthly means?). This approach seems to introduce a discrepancy between how you treat the model data (for which you were not able to do such a screening) versus the observations, and hence could be a partial explanation of why the model and observations don't agree with each other.

P11L21-23 Again I'm confused. Why is the threshold here 150 and not 100 anymore as mentioned earlier?

Figures 3 and 4, also supplementary figures. I suggest plotting the difference between these fields as well, since it is difficult to compare these plots quantitatively.

Correlations and means in Table 3 and 4 should be calculated for IAGOS-HR as well.

Please double-check your language once more before publication. A few issues are listed below, but this is not a comprehensive list of corrections. . .

P1L5 If a few. . . → Even though a few. . . P1L9 This argument is generally only true for archives of model-intercomparison studies, many institutions save their model fields on a daily or 10-hourly basis. Suggest a corresponding qualifier for this statement. P1L10 mapped at → mapped onto, or do you mean sampled at? P1L10 by MOCAGE CTM. . . of CCMI phase-1 → by the MOCAGE CTM. . . of the CCMI phase-1 P1L14 'Good correlations' → 'Good agreement', you don't calculate correlations. . . P2L4 'only due to . . .' → 'arising from. . .' P2L10 'on the processes' → 'in the processes' P2L12 'the main goal [. . .] lies on the reduction. . .' → 'the main goal [. . .] is the reduction. . .' P2L12 since you indicate the projects of ACCMIP and CCMVal, you should also do so for CCMI, which is IGAC/SPARC. Same on P5 Section 2.2 on CCMI. P2L18/19 How is it achieved? Recommend adding in a short explainer for 'SD' being 'nudged to observed meteorology' or 'specified dynamics' already here. P4L14 that studied → who studied

P4L15 what do you mean by quasi-totality? P6L19 I don't understand what the eccentricity of measurements refers to, please clarify. P6L22 Not clear what this refers to, I assume this approach? P6L25 of MOCAGE simulation → of the MOCAGE simulation P8L15-17 clarify sentence, did you mean ... has a negligible impact on the distribution of the IAGOS data from the cruise altitude onto the model vertical grid? P9L1 in this vicinity → in its vicinity to clarify that you refer back to the grid point. P9 equation 3: the multiplication sign looks formatted in a funny way, suggest to remove. P9L8 suggest to reorder sentence for clarification to → ... on the MOCAGE vertical levels considered which span level 28 to 22 and correspond to... P9L11 Do you really mean to say that all observations on levels 27 and 28 correspond to ascent or descents of the aircraft? P9L15 This is only true if the observations going into a grid box of your IAGOS-DM is representative for the monthly mean. This will depend on the number of measurements (your N) and the temporal sampling (distributed evenly over the month versus other sampling). P10L12 inconvenient → inconvenience P18L16 be more specific for the general reader → tropopause transition layer P20L27-29 inter-regional averages is not a valid expression. Cross-regional perhaps... but maybe better an average calculated over all regions... P20L28 as they are similar with zonal averages... → as they are similar to the zonal averages...

Additional References

Brunner, D., et al. (2003), An evaluation of the performance of chemistry transport models by comparison with scientific aircraft observations. Part 1: Concepts and overall model 2076 performance, Atmos. Chem. Phys., 3, 1609–1631.

Brunner, D., et al. (2005), An evaluation of the performance of chemistry transport models. Part 2: Detailed comparison with two selected campaigns, Atmos. Chem. Phys., 5, 107–129.

Hegglin, M.I., Gettelman, A., Hoor, P., Krichevsky, R., Manney, G.L., Pan, L.L., Son, S.W., Stiller, G., Tilmes, S., Walker, K.A. and Eyring, V., 2010. Multimodel assessment

of the upper troposphere and lower stratosphere: Extratropics. Journal of Geophysical Research: Atmospheres, 115(D3).

Orbe, Clara, David A. Plummer, Darryn W. Waugh, Huang Yang, Patrick Jöckel, Douglas E. Kinnison, Béatrice Josse et al. "Description and Evaluation of the specified-dynamics experiment in the Chemistry-Climate Model Initiative." Atmospheric Chemistry and Physics 20, no. 6 (2020): 3809-3840.

Tilmes, S., et al. (2010), An aircraft based upper troposphere lower stratosphere O3, CO, and H2O climatology for the Northern Hemisphere, J. Geophys. Res., 115, D14303, doi:10.1029/2009JD012731.

---

## Referee Comment (RC2) · Anonymous Referee #2 · 31 Dec 2020

The manuscript presents a new method for comparing the IAGOS observations of O3 and CO taken from in-service aircraft with gridded monthly average fields from chemistry climate models (CCMs) and applies the new method to outputs from the MOCAGE Chemistry Transport Model. The method involves aggregating the in-situ observations, taken along the flight path of the aircraft with a time resolution of less than a minute, to monthly averages at the horizontal and vertical resolution of the model. The method is of considerable interest because collaborative multi-model intercomparison projects frequently request monthly average fields. While requesting monthly average

fields makes the size and complexity of the data request more manageable, comparing monthly average fields with in-situ observations is not straightforward. And the situation is even more complex in the upper troposphere and lower stratosphere due to the strong gradients in chemical species that exist across the tropopause, which is itself highly variable in space and time. By attempting to bridge the gap between the highly time and space resolved IAGOS measurements with the widely available monthly average fields from CCMs the paper represents a nice advance on our understanding of how to assess models.

The paper is relatively well written, though I have noted a few instances where I had difficulty understanding the discussion. In general my comments are minor, detailed below, but I would like to raise one point that I found missing from the discussion of the results. A much better way to compare the models with the IAGOS observations would be to have highly time resolved instantaneous model outputs that are more directly comparable with the in-situ observations. This comparison would not be perfect because of errors in the model that arise from a whole host of different reasons that have been discussed at length in the literature. But, by and large, and particularly so for multi-model intercomparison projects, the large data volumes required to save high frequency output makes that type of comparison difficult and we must work with monthly average data. In addition to the usual list of reasons for model biases, now there is the added factor of having averaged the model fields in time. The authors have done a great job of exploring this effect with the observations by comparing IAGOS-HR with IAGOS-DM, but the discussion of the results does not include much mention of the effect of time averaging. It became a bit confusing when the discussion of the comparison of IAGOS-HR and IAGOS-DM zeroed in on the effect of mis-classification of points in either the UT or LS (see the comment on Page 19, Lines 4 – 6) and ignored the effect of time averaging. In the vicinity of the tropopause, over the course of a month, a particular model grid point will sometimes be in the stratosphere and sometime in the troposphere so the monthly average will reflect both of these influences. At least to my mind, this effect is an important part of the problem comparing monthly-average fields

with in-situ observations but I do not find much discussion of this facet of the problem in the manuscript.

Minor comments:

Page 3, Line 28: When discussing the available frequency of model output to compare against the IAGOS data, the manuscript states 'the 3D outputs from the REF-C1SD simulations, which are monthly averages.' The fact that monthly average fields are the most commonly available output is not necessarily part of the REF-C1SD specification, but more of a result of asking for a large amount of data from a number of models participating in a multi-model intercomparison. The text here should be more general and refer to the fact that monthly average fields are the most commonly available type from multi-model intercomparisons.

Page 4, Lines 7 – 10: In discussing other available methods for comparing in situ measurements with model data, the authors state that interpolating neighbouring measurement points onto each gridpoint would be computationally expensive for the IAGOS data because it requires keeping track of a large number of measurement locations each month. But when the methodology is described in Section 3 it would seem to me that the proposed method also requires keeping track of the discrete locations of a large number of observations. I believe I see the idea the authors are trying to convey – that a month of observation data at the original locations must be collected up before interpolating on to the grid point - but I would suggest rewording this point.

Page 7, Line 4: The phrase 'measured a mixing ratio Cobs(X) for an X species' might be better as 'measured a mixing ratio Cobs(X) for species X'

Page 8, Line 3: Where it is written 'increasing linearly with the distance between the measurement point and the (i, j, k) grid point.', I think should be 'decreasing linearly with the distance'.

Page 11, Line 1: I find the phrase 'before deriving monthly means in the two layers'

a bit confusing because I am not familiar with the analysis performed in Cohen et al. (2018) and the earlier discussion of model layers. I assume the two layers are the UT and LS and the analysis is done separately for each region?

Page 11, Line 23: might be missing a word (such?) in 'account, [such] as Western North America and Siberia.'

Page 14, Lines 13-14: 'With respect to the 1:1 line, levels 25 and 26 are characterized by an overestimation of the lower part of the O3 distribution (< 120 ppb) and by an underestimation of the higher part.' This sentence is referring to the pronounced underestimation of ozone between 150 ppb and 250 ppb shown in Figure 5 for P∼255 and P∼285 hPa, and from Figure 3 this looks to be related to an underestimation of ozone at high latitudes. Do the authors have any ideas why ozone in this one region seems to be underestimated to such a large degree? I will note that from the figures in the Appendix the underestimation does seem to be most extreme in the summer months.

Page 18, Lines 17 – 18: In referring to the IAGOS-DM data separated into the UT and LS regions ('In contrast, IAGOS-DM refers to the new product presented in this paper, i.e. the IAGOS data distributed on the model's grid, directly comparable to the simulation.') it might be helpful to the reader to remind them that the data is assigned into either the UT and LS based on the monthly average PV at each model grid point.

Page 19, Lines 4 – 6: Here it is stated ' In other words, by using the monthly mean PV from the simulation, some of the IAGOS measurement points may be attributed to the LS while being in the UT (or in the tropopause layer) and vice-versa.'. Here it sounds as though for the IAGOS-DM dataset individual IAGOS observations are being assigned to either the UT or LS based on their position relative to the monthly average tropopause. I had thought the IAGOS-DM was constructed and then the individual monthly-average values on model grid points were assigned to either the UT or LS (Page 11, Lines 4 – 8)? In this case, a particular monthly average value at a particular

model grid point may be affected by a mixture air from both the UT and LS. This is not necessarily a bad thing, as the model monthly averages that are being analyzed will have a similar problem.

Page 19, Figure 7. There are a lot of lines on each panel of Figure 7 (likewise for Figure 8) and making a clear comparison between IAGOS-HR, IAGOS-DM and MOCAGE-M is not easy. I am sympathetic to the need to condense graphics to avoid figures with 20 panels, but I was wondering if the authors would consider adding figures to the appendix that directly compare the three datasets for each of the eight regions? For each region there would be one panel showing the annual cycle from IAGOS-HR, IAGOS-DM and MOCAGE-M for the LS and and a second panel for the UT.

---

## Author Comment (AC1) · 15 Feb 2021

We are thankful to the reviewers for their positive and accurate feedback on our study, and for the improvements they allowed us to perform.

Before answering to the comments, we have to point out the changes made in Table 3, in Section 4.2, since the original manuscript version. They are due to a correction of an error in our method that was rightly spotted by Referee 1. More precisely, it concerns the additional mask based on the observed ozone values. Our mistake consisted in filtering out the individual measurement points if their corresponding ozone value was not consistent enough with the PV, instead of filtering out whole monthly grid cells. This correction results in a stronger loss of data and in the removal of several seasonal cycles from the figures, because they became incomplete. Nevertheless, it did not change the main conclusions.

Note also that since the submission of the GMDD version, we have decided to add the name of the methodology (Interpol-IAGOS) into the title of the revised version of the paper. There is also a change in the scatterplot in Fig. B3, at level 25 where the previous version showed two extreme outliers in observed CO (350 and 510 ppb) during summer, corresponding to two grid cells above Alaska. This removal has been explained in the figure legend too.

For the responses, the text from the reviewers is in blue. Our answers are in black, and the changes proposed for the revised manuscript are in italic (black for modified sentences, grey for unchanged sentences that have been pasted here in order to remind the context). In the revised manuscript, the main changes are shown in blue.

**The paper is relatively well written, though I have noted a few instances where I had difficulty understanding the discussion. In general my comments are minor, detailed below, but I would like to raise one point that I found missing from the discussion of the results. A much better way to compare the models with the IAGOS observations would be to have highly time resolved instantaneous model outputs that are more directly comparable with the in-situ observations. This comparison would not be perfect because of errors in the model that arise from a whole host of different reasons that have been discussed at length in the literature. But, by and large, and particularly so for multi-model intercomparison projects, the large data volumes required to save high frequency output makes that type of comparison difficult and we must work with monthly average data. In addition to the usual list of reasons for model biases, now there is the added factor of having averaged the model fields in time. The authors have done a great job of exploring this effect with the observations by comparing IAGOS-HR with IAGOS-DM, but the discussion of the results does not include much mention of the effect of time averaging. It became a bit confusing when the discussion of the comparison of IAGOS-HR and IAGOS-DM zeroed in on the effect of mis-classification of points in either the UT or LS (see the comment on Page 19, Lines 4 – 6) and ignored the effect of time averaging. In the vicinity of the tropopause, over the course of a month, a particular model grid point will sometimes be in the stratosphere and sometime in the troposphere so the monthly average will reflect both of these influences. At least to my mind, this effect is an important part of the problem comparing monthly-average fields with in-situ observations but I do not find much discussion of this facet of the problem in the manuscript.**

We fully agree that this explanation is clearer than the one we used in the paper. Actually, these explanations are closely linked (we would even say equivalent), but maybe our meaning was not clear. What we meant was that with a monthly average, for instance in the UT, we include grid points from undesirable stratospheric air masses. This leads to a mis-classification of a non-negligible part of the individual measurements. We changed the sentence, as below:

*In other words,* **the effect of time averaging leads to a mis-classification of a non-negligible part of the individual measurements. For a given layer, it introduces a bias due to unexpected mixing with another layer.**

And in the conclusion too:
*The use of a monthly mean PV field and the ~800 m vertical resolution in the UTLS of MOCAGE onto which IAGOS observations are projected automatically result in an artificial increase of stratosphere-troposphere exchange.* **It is explained by the fact that the grid cells in the vicinity of the tropopause are crossed by both tropospheric and stratospheric air masses in the course of a month.** *It results in a decreased vertical gradient between UT and LS. Nevertheless, the seasonal maxima and minima become less clear but remain visible in IAGOS-DM with respect to IAGOS-HR.*

**Minor comments:**

**Page 3, Line 28: When discussing the available frequency of model output to compare against the IAGOS data, the manuscript states 'the 3D outputs from the REF-C1SD simulations, which are monthly averages.' The fact that monthly average fields are the most commonly available output is not necessarily part of the REF-C1SD specification, but more of a result of asking for a large amount of data from a number of models participating in a multi-model intercomparison. The text here should be more general and refer to the fact that monthly average fields are the most commonly available type from multi-model intercomparisons.**

Note that Reviewer 1 made the same remark. The sentence has been modified, as shown below:

*However, these comparisons were led using frequent simulation outputs.* **Although the high frequency is necessary for their approach to separate accurately the air masses into different categories, it is not adapted to the assessment of monthly averaged fields used in multi-model intercomparisons.**

**Page 4, Lines 7 – 10: In discussing other available methods for comparing in situ measurements with model data, the authors state that interpolating neighbouring measurement points onto each grid point would be computationally expensive for the IAGOS data because it requires keeping track of a large number of measurement locations each month. But when the methodology is described in Section 3, it would seem to me that the proposed method also requires keeping track of the discrete locations of a large number of observations. I believe I see the**

**idea the authors are trying to convey– that a month of observation data at the original locations must be collected up before interpolating on to the grid point - but I would suggest rewording this point.**

We agree with your comment. It was not clear. In contrast to our method, the method that we were citing here (notably in New et al., 2000) calculates linear combinations between measurement points, and thus require to keep track of a large amount of observations simultaneously, whereas for a given variable, our method requires to store only 2 quantities per grid cell: the sum of all the weights and the weighted sum of the current variable (ozone or CO). We brought precision to the text (as pasted below) to make it clearer:

*Some of them consist in calculating **a linear combination** from the neighbouring measurements points onto each gridpoint (e.g. New et al., 2000). **However, it requires to store simultaneously the information of all the measurement locations, and during a whole month.** It is thus convenient for measurements with regular locations as surface stations, whereas their use on the IAGOS database would be expensive computationally as well.*

From now on, the recommendations let without a response in this document have all been included in the manuscript.

**Page 7, Line 4: The phrase 'measured a mixing ratio Cobs(X) for an X species' might be better as 'measured a mixing ratio Cobs(X) for species X'**

**Page 8, Line 3: Where it is written 'increasing linearly with the distance between the measurement point and the (i, j, k) grid point.', I think should be 'decreasing linearly with the distance'.**

We really meant that the alpha, beta and gamma coefficients increased with the distance with the (i,j,k) gridpoint. In contrast, the corresponding weight defined by the product (1-alpha)*(1-beta)*(1-gamma) decreases with the distance. We have changed the wording to make it clearer:

*[…] a normalized **scalar** is then computed for each dimension (coefficients alpha, beta, gamma), increasing linearly with the distance between the measurement point and the (i, j, k) grid point.*

This change was also done in the legend of the corresponding figure.

**Page 11, Line 1: I find the phrase 'before deriving monthly means in the two layers' a bit confusing because I am not familiar with the analysis performed in Cohen et al. (2018) and the earlier discussion of model layers. I assume the two layers are the UT and LS and the analysis is done separately for each region?**

Exactly. This part of the text has been reworded. We added the word "separately" in the sentence below to make it explicit:

*A second part of this assessment targets the behaviour of the model in the UT and the LS separately.*

**Page 11, Line 23: might be missing a word (such?) in 'account, [such] as Western North America and Siberia.'**

**Page 14, Lines 13-14: 'With respect to the 1:1 line, levels 25 and 26 are characterized by an overestimation of the lower part of the O3 distribution (< 120 ppb) and by an underestimation of the higher part.' This sentence is referring to the pronounced underestimation of ozone between 150 ppb and 250 ppb shown in Figure 5 for P~255 and P~285 hPa, and from Figure 3 this looks to be related to an underestimation of ozone at high latitudes. Do the authors have any ideas why ozone in this one region seems to be underestimated to such a large degree? I will note that from the figures in the Appendix the underestimation does seem to be most extreme in the summer months.**

These biases have been investigated, some reaction constants have been updated and for long CCMI-type simulations, in particular the CCMI phase-2, MOCAGE is now run in 60 levels (up to 0.1 hPa), giving a better representation of ozone at global scale.

Precision has been added into this paragraph.
*[…] levels 25 and 26 are characterized by an overestimation of the lower part of the O3 distribution (< 120 ppb) and by an underestimation of the higher part*, more pronounced *during boreal summer according to Fig. B3 in Appendix.*
*A possible reason is that the summertime tropopause altitude in these regions can be overestimated by the model, or that the vertical stability is underestimated. These biases have been largely improved with the more recent version of MOCAGE used to run CCMI phase-2 simulations.*

**Page 18, Lines 17 – 18: In referring to the IAGOS-DM data separated into the UT and LS regions ('In contrast, IAGOS-DM refers to the new product presented in this paper, i.e. the IAGOS data distributed on the model's grid, directly comparable to the simulation.') it might be helpful to the reader to remind them that the data is assigned into either the UT and LS based on the monthly average PV at each model grid point.**

We replaced the expression "directly comparable to the simulation" by the more precise formulation you proposed.

**Page 19, Lines 4 – 6: Here it is stated 'In other words, by using the monthly mean PV from the simulation, some of the IAGOS measurement points may be attributed to the LS while being in the UT (or in the tropopause layer) and vice-versa.'. Here it**

**sounds as though for the IAGOS-DM dataset individual IAGOS observations are being assigned to either the UT or LS based on their position relative to the monthly average tropopause. I had thought the IAGOS-DM was constructed and then the individual monthly-average values on model grid points were assigned to either the UT or LS (Page 11, Lines 4 – 8)? In this case, a particular monthly average value at a particular model grid point may be affected by a mixture air from both the UT and LS. This is not necessarily a bad thing, as the model monthly averages that are being analyzed will have a similar problem.**

We fully agree with this comment. As indicated in our response to Reviewer 1, a mistake has been corrected in our data processing. The clarification on the mixing between UT and LS air masses due to the PV monthly averages has already been added, in answer to a previous comment.

**Page 19, Figure 7. There are a lot of lines on each panel of Figure 7 (likewise for Figure 8) and making a clear comparison between IAGOS-HR, IAGOS-DM and MOCAGE-M is not easy. I am sympathetic to the need to condense graphics to avoid figures with 20 panels, but I was wondering if the authors would consider adding figures to the appendix that directly compare the three datasets for each of the eight regions? For each region there would be one panel showing the annual cycle from IAGOS-HR, IAGOS-DM and MOCAGE-M for the LS and a second panel for the UT.**

We agree with this suggestion. A whole set of new graphics has been added to the Appendix. They are mentioned in the text in Sect. 4.2, second paragraph:

*The comparison between the two IAGOS products in matter of seasonal cycles is proposed in Figs. 7 and 8, respectively for $O_3$ and CO. They are shown with their corresponding interannual variability (IAV), defined as a year-to-year standard deviation.* **For complementary information, a more exhaustive representation is proposed in Figs. C1 and C2 in the Appendix showing the results with each region in a distinct panel.** *In Fig. 7, both IAGOS versions show a summertime $O_3$ maximum […].*

---

## Author Comment (AC2) · 15 Feb 2021

We are thankful to the reviewers for their positive and accurate feedback on our study, and for the improvements they allowed us to perform.

Note that since the submission of the GMDD version, we have decided to add the name of the methodology (Interpol-IAGOS) into the title of the revised version of the paper. There is also a change in the scatterplot in Fig. B3, at level 25 where the previous version showed two extreme outliers in observed CO (350 and 510 ppb) during summer, corresponding to two grid cells above Alaska. This removal has been explained in the figure legend too.

For the responses, the text from the reviewer is in blue, our answers are in black, and the changes proposed for the revised manuscript are in italic (black for modified sentences, grey for unchanged sentences that have been pasted here in order to remind the context). In the revised manuscript, the main changes are shown in blue.

**Major comments:**

**P1L12-14 The sentence 'as a first application...' seems confusing at this place. The authors should try to clarify (or attempt a better logical flow of) their description of the evaluation methodology they introduce. First you interpolate IAGOS onto the model grid, then you mask the model at the available IAGOS grid points, then you attempt the evaluation of the model only, right?**

It is right. We suggest a new formulation, as pasted below:

*The present study proposes a method to compare this large IAGOS data set to long-term simulations used for chemistry-climate studies. As a first application, the REF-C1SD simulation generated by MOCAGE CTM in the framework of the CCMI phase-I has been evaluated during the 1994–2013 period for ozone ($O_3$) and the 2002–2013 period for carbon monoxide (CO).* **The concept of the new comparison software proposed here (named Interpol-IAGOS) is to project all IAGOS data onto the 3D grid of the model with a monthly resolution, since generally the 3D outputs provided by chemistry-climate models for multi-model comparisons are archived as monthly means. This provides a new IAGOS data set (IAGOS-DM) mapped at the model's grid and time resolution. To get a model data set consistent with IAGOS-DM for the comparison, a subset of the model's outputs is created (MOCAGE-M) by applying a mask that retains only the model data at the available IAGOS-DM grid points.** *First, climatologies […]*

**The authors have done a commendable job in summarising references in the introduction that use satellite observations for model comparisons using SD simulations and IAGOS particularly. However, there are earlier studies that are also highly relevant to the here proposed evaluation methodology and hence should be mentioned as well. Brunner et al. (2003, 2005), Tilmes et al. (2010), and Hegglin et al. (2010) all use aircraft (and satellite) observations to**

**evaluate CTMs and free-running CCMs in the UTLS, respectively. At least a qualitative discussion of how your findings improve (or whether they yield similar results) compared to the latter two studies in particular is needed, given they use tropopause-based coordinates to evaluate CO and O3 in the upper troposphere and lower stratosphere to evaluate CCMs, an equivalent methodology as proposed here but applied to free-running simulations.**

We agree that these studies are relevant. Hegglin et al. (2010) and Tilmes et al. (2010) are now mentioned in the introduction as follow:

*In addition to ozonesondes, aircraft measurements from different campaigns were used in the evaluation of the REF-C1SD simulations from the model CESM1 CAM4-Chem (Tilmes et al., 2016).* *Aircraft campaigns have already proven their usefulness in assessing models in the UTLS. Tilmes at al. (2010) built a climatology of $O_3$ and CO in the tropics, subtropics and extratropics by gathering a wide set of aircraft campaigns from 1995 until 2008. Hegglin et al. (2010) used this data set to assess the eighteen CCMs participating in CCMVal-2 in the extratropical lower stratosphere using several diagnostics. For instance, the seasonal cycles derived at 100 and 200 hPa highlighted a relatively good reproduction of ozone behaviour in the lower stratosphere, and allowed to identify an overestimation of the transport from the tropics at 100 hPa and across the tropopause at 200 hPa. However, their conclusion also highlighted the limitations in space and time of the in situ observations, especially in the upper troposphere.*

The papers Brunner et al. (2003 and 2005) are mentioned in the next paragraph, as below:

*[…] especially in the northern mid-latitudes from Western North America to East Asia.* *Brunner et al. (2003; 2005) combined research aircraft measurements with the first years of the IAGOS-MOZAIC database (1995–1998) to assess five CTMs and two CCMs.* *Gaudel et al. (2015) performed an evaluation of the MACC (Monitoring Atmospheric Composition and Climate) reanalysis over Europe during 2003–2010, using IAGOS $O_3$ and CO measurements.* *However, these comparisons used frequent simulation outputs. Although the high frequency is necessary for their approach to separate accurately the air masses into different categories, it is not adapted to the assessment of monthly averaged fields used in multi-model intercomparisons.*

**While you may argue that your method has the advantage to have near-coincident measurements for the model evaluation, the problem with the SD-simulations could be that certain nudging methodologies may lead to introducing noise (and hence too much mixing) especially visible around the tropopause (see discussion in Orbe et al. 2020). This issue indeed could help your interpretation of your findings (too much/too little ozone in the UT and LS, respectively, and vice-versa for CO).**

Thanks for this reference. Indeed, it would be an additional source of bias for a nudged CCM evaluation. It does not concern our results here because we are assessing an offline CTM directly forced by meteorological reanalyses. However, since we encourage the use of our method to the whole CCMI community, it is worth noting this issue while discussing the perspectives.

*The present methodology could be easily applied to CCMI REF-C1SD simulations from other models, both for an intermodel comparison and for assessing CCMI products against IAGOS database, notably intermodel-averaged fields. To a greater extent, it can be used on a wide range of long-term simulations including CCMs free runs in order to perform climatological comparisons.* **It could also be used to investigate the impacts of nudging methodologies used in CCMs, as Orbe et al. (2020) showed that nudging increased the discrepancies between the CCMs in several dynamical variables.** *[…]*

**P4L3 ff. The statement that it is not possible is an exaggeration. As shown by Brunner et al. (2003, 2005) interpolation onto the measurement space can yield indepth insight into model behaviour and even though the data may not be available from the multi-model intercomparison data archives, higher resolution that would be needed for this task could be obtained from the modelling centers.**

This part of the text has been modified. The new sentences are:

*To compare REF-C1SD simulations against IAGOS data, interpolating the simulation outputs onto the high-resolution observations would be very expensive computationally, and is not required because our study is not focused on processes but on climatologies.* *Alternatively, the comparison could be performed after mapping the high resolution IAGOS data on the model grid, on a monthly basis.*

**In fact, it would be important to prove for a methodology paper as you have presented here that your claim of the gridded IAGOS data being representative of the monthly mean is true. You could do this by sub-sampling your model according to IAGOS measurement locations (including weighting and interpolation on measurement location), then compare the mean of this sub-sampled model field to the IAGOS-based masked model field. Agreement between the two will tell you the accuracy of the assumption underlying your methodology. This extra step in my eyes would be needed to make this methodology paper a sound contribution to model evaluation efforts, and without it the question remains whether differences in the model-measurement comparison arise from comparing apples to oranges as opposed to true model deficiencies.**

The representativeness of the months by the IAGOS observations is a key point and we are thankful for this comment. We performed the test on a 5-year subsample

chosen from Jan. 2003 until Dec. 2007, an uninterrupted measurement period for both ozone and CO.
More precisely, we used the model daily output, applied the IAGOS mask at the daily timescale and generated new monthly outputs. The MOCAGE-M product derived from this processing is called MOCAGE-M-day hereafter. Please find below the tables summarizing the comparative statistics between the two MOCAGE-M products:

- For the seasonal climatologies:

```
O3 DJF   r= 0.97   MNMB= -0.012  FGE= 0.094  N_cells= 10727
O3 MAM  r= 0.97   MNMB= 0.004   FGE= 0.087  N_cells= 10403
O3 JJA    r= 0.97   MNMB= 0.002   FGE= 0.057  N_cells= 10063
O3 SON  r= 0.97   MNMB= -0.016   FGE= 0.073  N_cells= 10395
O3 ANN  r= 0.99   MNMB= -0.007  FGE= 0.055  N_cells= 10978
CO DJF   r= 0.94   MNMB= -0.001 FGE= 0.071  N_cells= 10358
CO MAM r= 0.93   MNMB= 0.015   FGE= 0.058  N_cells= 10292
CO JJA    r= 0.97   MNMB= 0.017   FGE= 0.045  N_cells= 9884
CO SON  r= 0.95   MNMB= 0.007   FGE= 0.054  N_cells= 10306
CO ANN  r= 0.98   MNMB= 0.009   FGE= 0.037  N_cells= 10706
```

- For the seasonal cycles:

|     |     | IAGOS-HR | IAGOS-DM | MOCAGE-M-day | MOCAGE-M | $r_{day\ vs\ mth}$ |
|-----|-----|----------|----------|--------------|----------|--------------------|
| O3  | LS  | 285 +/- 26 | 222 +/- 31 | 214 +/- 27 | 207 +/- 17 | 0.98 |
|     | UT  | 62 +/- 4  | 71 +/- 6  | 95 +/- 6  | 97 +/- 3  | 0.97 |
| CO  | LS  | 53 +/- 5  | 68 +/- 7  | 70 +/- 6  | 71 +/- 4  | 0.96 |
|     | UT  | 110 +/- 9 | 108 +/- 8 | 115 +/- 13 | 113 +/- 8 | 0.96 |

The results show a good correlation between the two MOCAGE products, both for the seasonal cycles (temporal correlation) and the seasonal climatologies (spatial correlation). The MNMBs are low (less than 0.02), and associated to a low error (FGE less than 0.1) for the climatologies. The differences in the seasonal cycles are also low for the mean values, and only the interannual variability shows a non-negligible increase between MOCAGE-M and MOCAGE-M-day. This can be explained by the use of monthly means for MOCAGE-M fields that is expected to be smoother than the daily means of MOCAGE-M-day fields. Note that the numbers given above corresponds to 5 years of data while those given in Table 3 in the paper are for the whole IAGOS data set. This is why they are slightly different.

We added a short description of this test and its results at the end of Section 3.2:
*[…] The latter point [the representativity of the month] has been tested (not shown) on a 5-year subsample using the simulation daily outputs instead of monthly outputs. It consisted in comparing MOCAGE-M to a test product derived by*

*calculating monthly averages from the daily outputs and applying a mask based on the IAGOS daily sampling. The similar results obtained between MOCAGE-M and the test product showed that the MOCAGE-M monthly means could be considered as representative of the month.*

**On a similar note, it would also be good to see what the benefit of the weighted gridding versus a gridding of the observations without the weighting function would be (equivalent to what you have done in Section 4.2 for IAGOS-DM and IAGOS-HR). This is important in order to convince the reader that the weighted gridding process is in fact worth the effort. I would expect that the interpolation/weighting is less needed for the maps on different pressure levels, unlike the result of the tropopause-based evaluation in Section 4.2, but I could be wrong.**

We agree that these arguments are important to justify the "complexity" of our data processing. New statistics on the whole IAGOS data set have been added into the tables, giving the scores between IAGOS-DM and MOCAGE-M as before, but also between the data sets without weighting (IAGOS-DM-noW and MOCAGE-M-noW, with the "noW" suffix standing for "no weighting").
The statistical results are the following:

- climatologies: Tables 1 and 2 have been merged together in the revised version.

**Table 1.** Seasonal and annual metrics synthesizing the assessment of the simulated $O_3$ and CO climatologies by IAGOS-DM, gathering all the vertical grid levels as in Fig. 6. From left to right: Pearson's correlation coefficient (r), modified normalized mean bias (MNMB), fractional gross error (FGE) and the sample size ($N_{cells}$).

| Species | Season | r | MNMB | FGE | $N_{cells}$ |
|---|---|---|---|---|---|
| $O_3$ | DJF | 0.95 | 0.144 | 0.190 | 12,723 |
| | MAM | 0.94 | -0.033 | 0.163 | 12,622 |
| | JJA | 0.92 | -0.169 | 0.280 | 12,587 |
| | SON | 0.89 | 0.027 | 0.180 | 13,073 |
| | ANN | 0.95 | -0.012 | 0.150 | 13,062 |
| CO | DJF | 0.77 | 0.098 | 0.171 | 12,081 |
| | MAM | 0.82 | 0.098 | 0.157 | 11,623 |
| | JJA | 0.75 | -0.011 | 0.130 | 11,618 |
| | SON | 0.75 | 0.024 | 0.126 | 12,467 |
| | ANN | 0.83 | 0.049 | 0.112 | 12,482 |

In order to compare our method with the "noW" method, we have added a new table (the new Table 2). To make a fair comparison, we recalculated the scores of MOCAGE-M versus IAGOS-DM using the same spatial sampling as in MOCAGE-M-noW versus IAGOS-DM-noW. This is the reason why the MOCAGE-M versus IAGOS-DM are slightly different from the new Table 1.

**Table 2.** Same metrics as in Table 1, showing the scores derived from the comparison between IAGOS-DM and MOCAGE-M (first three columns), then between IAGOS-DM-noW and MOCAGE-M-noW. All the scores reported in this table are based on the IAGOS-DM-noW sampling size $N_{cells, noW}$ (last column).

| Species | Season | r | MNMB | FGE | $r_{noW}$ | $MNMB_{noW}$ | $FGE_{noW}$ | $N_{cells, noW}$ |
|---|---|---|---|---|---|---|---|---|
| $O_3$ | DJF | 0.95 | 0.133 | 0.182 | 0.91 | 0.076 | 0.219 | 10,404 |
| | MAM | 0.95 | -0.042 | 0.165 | 0.91 | -0.101 | 0.260 | 10,293 |
| | JJA | 0.92 | -0.190 | 0.287 | 0.88 | -0.217 | 0.354 | 10,252 |
| | SON | 0.90 | 0.009 | 0.177 | 0.84 | -0.021 | 0.232 | 10,684 |
| | ANN | 0.95 | -0.021 | 0.153 | 0.92 | -0.073 | 0.224 | 11,667 |
| CO | DJF | 0.74 | 0.120 | 0.173 | 0.67 | 0.182 | 0.243 | 9,298 |
| | MAM | 0.79 | 0.105 | 0.156 | 0.72 | 0.186 | 0.247 | 8,896 |
| | JJA | 0.74 | -0.013 | 0.122 | 0.63 | 0.042 | 0.168 | 8,977 |
| | SON | 0.72 | 0.035 | 0.121 | 0.68 | 0.077 | 0.166 | 9,608 |
| | ANN | 0.81 | 0.056 | 0.110 | 0.71 | 0.113 | 0.178 | 10,735 |

- seasonal cycles: Here again, a new table (Table 4) has been added in the revised version to show the results derived from the "noW" method. Since we have included the results from the comparison with IAGOS-HR (as suggested by R1 later) in Table 3, we preferred to create a new separate table for the "noW" statistics.

**Table 3.** Cross-regional averages derived from the seasonal cycles. Yearly mean values and interannual variabilities are shown for IAGOS-HR, IAGOS-DM and MOCAGE-M, along with the correlation coefficient between MOCAGE-M and the two IAGOS versions.

| Species | Layer | IAGOS-HR | IAGOS-DM | MOCAGE-M | $\bar{r}_{IAGOS-HR}$ | $\bar{r}_{IAGOS-DM}$ |
|---|---|---|---|---|---|---|
| $O_3$ (ppb) | LS | $287 \pm 33$ | $222 \pm 36$ | $203 \pm 23$ | 0.70 | 0.84 |
| | UT | $62 \pm 6$ | $72 \pm 9$ | $97 \pm 5$ | 0.14 | 0.35 |
| CO (ppb) | LS | $51 \pm 5$ | $66 \pm 8$ | $72 \pm 5$ | 0.28 | 0.31 |
| | UT | $104 \pm 11$ | $101 \pm 11$ | $109 \pm 8$ | 0.63 | 0.58 |

**Table 4.** As in Table 3 for IAGOS-DM and MOCAGE-M derived without weighting (IAGOS-DM-noW and MOCAGE-M-noW).

| Species | Layer | IAGOS-DM-noW | MOCAGE-M-noW | $\bar{r}_{noW}$ |
|---|---|---|---|---|
| $O_3$ (ppb) | LS | $235 \pm 39$ | $181 \pm 20$ | 0.84 |
| | UT | $72 \pm 9$ | $98 \pm 5$ | 0.34 |
| CO (ppb) | LS | $63 \pm 8$ | $77 \pm 5$ | 0.25 |
| | UT | $101 \pm 11$ | $110 \pm 9$ | 0.54 |

Tables 2 and 4 clearly show that there is a change by applying a weighting function. Thus, it is necessary to use it in the proposed method.

Here is the list of the subsequent changes in the text.

- In the abstract:
*They [the seasonal cycles] are remarkably well-reproduced by the model for lower-stratospheric $O_3$ and also good for upper-tropospheric CO.* **Along this model evaluation, we also assess the improvements brought by the use of a weighting function in the method when projecting the IAGOS data onto the model grid compared to the scores derived in a simplified way. We conclude that** *the data projection onto the model's grid* **using a weighting function** *is a necessary step for a more accurate assessment, […].*

- At the end of the 3.2 subsection:

*In order to compare the observations and the model at the same locations and months, we apply a mask on the MOCAGE REF-C1SD simulation outputs that allows us to account only for the IAGOS-DM sampled grid points. The subsequent data set is named MOCAGE-M, the M letter referring to the mask. Thus, IAGOS-DM and MOCAGE-M data sets are fully consistent and can be used to make gridpoint-to-gridpoint and month-by-month comparisons.* **In order to test the advantages of the linear interpolation involving weighting factors, we also derive another product from the IAGOS database using a simplified method, i.e. by solely averaging the measurement points into the grid cells where they are located. This control product is named IAGOS-DM-noW hereafter, the -noW suffix standing for "no weighting". Since it changes the spatial sampling distribution, a new subsequent mask has to be applied to the MOCAGE grid to be consistent with the IAGOS-DM-noW product, called MOCAGE-M-noW hereafter.**

- At the end of the 4.1 subsection:

*Nevertheless, all MNMB and FGE are very low, showing good skills from the MOCAGE REF-C1SD simulation.*
**The comparison with the last columns shows a better agreement between the model and the observations when we apply the interpolation with the weighting factors. The $O_3$ correlation decreased to 0.84–0.92 compared to the 0.89–0.95 derived from our method, and the CO correlation dropped from 0.72–0.84 down to 0.61–0.77. The MNMB and the FGE show better scores for the "noW" products in all cases, except the $O_3$ MNMB in DJF. The general improvement of normalised biases, normalised errors and spatial correlations, compared to a simplified gridding method, shows that the use of a weighting function in our methodology provides a significant improvement for the model assessment.**

- In the 4.2 subsection:

*It is important to note that the seasonal cycles in IAGOS-DM generally show values comprised between the MOCAGE-M and the IAGOS-HR cycles*, **such as the yearly means in Table 2, especially in the LS where the mean $O_3$ bias drops from 87 ppb with IAGOS-HR down to 17 ppb with IAGOS-DM. The correlation in time also tends to be enhanced by the use of the IAGOS-DM product.** *It confirms that the representation derived from IAGOS-HR cannot be reached by a model with the typical REF-C1SD resolution, especially for CO in the LS, although some main characteristics mentioned above can still be used as criteria.* **Last, the comparison synthesized in Table 4 also shows a better consistency between model and observations when our method is applied, mainly in matter of biases in the LS. No significant change is observed in the UT.**

**P9L22 I am not clear why/how the shorter period for CO measurements compared to O3 matter here? After all, 10 or 20 years may result in not so different climatologies, in fact a 10-year climatology may be more representative than a 20-year one (as for ozone) since you will find quite some trends in the latter.**

What we mean is that for the climatologies, we sum up all the monthly measurement amounts, so we have to account in the choice of $N_{thres}$ for the fact that we will automatically have a lesser amount of measurements with a shorter period. But we agree that in terms of trends 10 years is representative.
We have clarified our explanation in the text:
*Accounting for the shorter CO measurement period compared to $O_3$,* **less measurements are needed to characterize the climatologies. Thus,** *the corresponding $N_{thres}$ threshold for this species is derived by applying a factor 0.6, leading to 60.*

**In any case, the determination of Nthres seems rather arbitrary. How did you determine it? Did you make a quantitative analysis by how much the mean would change if more (or less for that matter) than 100 samples were included in the mean?**

This reference threshold is justified by the figure below (Fig. AR1). This figure shows for ozone, the correlation and percentage of points filtered out as a function of $N_{thres, ref}$. At $N_{thres, ref}$ = 100, the mean r coefficient becomes substantially less sensitive to a "perturbation" (i.e. +/- 20) of $N_{thres, ref}$, whereas the proportion of filtered-out grid cells still increases. The 100 value is thus justified by the will to minimize both the sensitivity to the $N_{thres}$ number and the removal of sampled grid cells.

[Figure]

*Figure AR1: Sensitivity test of the model performances concerning ozone, with respect to the reference sampling threshold $N_{thres, ref}$. The solid line represents the geographical correlation averaged over the vertical grid levels in the UTLS and over the seasons, with the values reported on the left axis. The dashed line corresponds to the proportion of sampled grid cells that have been filtered out. Its values are indicated on the right axis.*

**Minor comments**

**P10L21 to P11L5-15 Please clarify/write more succinctly here that you use monthly PV fields to distinguish between UT and LS observations.**

This specification has been shortened. Please find the new description below:

*A second part of this assessment targets the behaviour of the model in the UT and the LS, separately. The diagnostics we use for this purpose are adapted from Cohen et al. (2018), based on Thouret et al. (2006) who used the potential vorticity (PV) fields from the ECMWF operational analysis to derive the tropopause pressure. In contrast with the latter studies, we define the tropopause layer with the monthly averaged PV fields from ERA-Interim, as used in the MOCAGE REF-C1SD simulation. A given grid point is then considered as belonging to the UT if its monthly PV is lower than 2 potential vorticity units (PVU), and to the LS if the PV is greater than 3 PVU.*

**Also, it is not clear to me how the additional screening is applied to the data to get rid of too high (low) values in the UT (LS) respectively (is it applied to the monthly means only or to the input data that are used to generate these monthly means?). This approach seems to introduce a discrepancy between how you treat the model data (for which you were not able to do such a screening) versus the observations, and hence could be a partial explanation of why the model and observations don't agree with each other.**

It is true, and we are thankful to Reviewer 1 to point out this mistake. Our purpose was actually to apply this screening to the monthly means – as justified by the use of the interannual variabilities from Cohen et al. (2018) and not the extreme values amongst the individual measurements – but we unintentionally followed the second option, i.e. point-to-point, which is wrong here.

This point is now explained as follows:
*We also filter out the grid boxes where this PV classification is not consistent with the* **mean** *observed O$_3$ mixing ratio, i.e. where the* **monthly** *O$_3$ level reaches 140 ppb in the UT and where it goes under 60 ppb in the LS. It avoids an additional bias based on errors in the dynamical field leading to unrealistic UT and LS attribution.*

From this, we reprocessed the results and updated the seasonal cycles figures, the tables and the values cited in the text. Of course, our previous comments in the present authors' response account for these updated results. It did not change our main results, but it is worth listing here the changes and modifications.

- Some years have been removed in the UT because of the subsequent decrease of sampling: now we filter out the whole grid cell if its mean ozone mixing ratio disagrees with the PV, whereas the previous erroneous method kept more grid cells by cancelling individual observations only. For some regions where the sampling was concentrated in a small pressure interval (mainly North Atlantic), the seasonal cycles were removed from the figures because the amount of available years dropped down. It is the case for ozone and CO in North Atlantic, and for CO in the West Mediterranean basin and in Siberia.

*Note that the monthly resolution of both PV and filtering leads to a lessened sampling in the UT in IAGOS-DM. In the North Atlantic region where aircraft trajectories describe a narrow altitude range, the resulting seasonal cycles were incomplete so that we chose to exclude* **this region** *from* **both** *figure***s**. *We applied the same treatment to CO in the UT above the West Mediterranean basin and Siberia, where the level of sampling during winter and spring (not shown) are insufficient to provide complete seasonal cycles.*

- Some additional variability has emerged from the reprocessing, mainly for Siberian Ozone in November and Northwest-American CO in June, while some has disappeared like Northeast-Asian CO in June, probably because of the loss of the year

2003 characterized by an intense summertime fire activity. The comments have been modified:

*We also note high Ozone values in* **November** *in the* **Siberian UT** *seen by IAGOS-DM only. It is linked to a strong positive anomaly* **in November 1997** *due to an upper-layer air mass that could not be differentiated to the UT, and weakly balanced by the average with too few other years.*

**P11L21-23 Again I'm confused. Why is the threshold here 150 and not 100 anymore as mentioned earlier?**

It is different here because we are now treating monthly means, at regional scale. The 100 threshold was applied in the context of the horizontal climatologies (averaged over the whole period), at grid-cell scale.

We added a sentence in order to prevent from confusions, at the end of the paragraph:
*In Cohen et al. (2018), the regional monthly means with less than 300 data were filtered out. [...]* *It is important to note that the sampling threshold mentioned in this paragraph concerns each monthly average within a regional time series, contrasting with the sampling threshold we use for the (multi-)decadal average on each grid cell in the horizontal climatologies.*

**Figures 3 and 4, also supplementary figures. I suggest plotting the difference between these fields as well, since it is difficult to compare these plots quantitatively.**

All the maps have been re-plotted, and the differences have been added to the figures. For the differences, each value is defined as the model bias normalized to the average between model and observations, as for the MNMB and the FGE, but expressed in %.

**Correlations and means in Table 3 and 4 should be calculated for IAGOS-HR as well.**

We agree. New columns have been added in Table 3 and commented but not in Table 4, because there is no numerical value represented in the latter.

**Please double-check your language once more before publication. A few issues are listed below, but this is not a comprehensive list of corrections…**
Thanks for this list of corrections. We have double-checked the language before re-submission.

From now on, all the Reviewer #1's suggestions let without a response below are small changes that have been integrated into the revised paper.

**P1L5 If a few… → Even though a few…**
**P1L9 This argument is generally only true for archives of model-intercomparison studies, many institutions save their model fields on a daily or 10-hourly basis. Suggest a corresponding qualifier for this statement.**

We reformulated the sentence:

*The concept of the new comparison software proposed here (so-called Interpol-IAGOS) is to project all IAGOS data onto the 3D grid of the model with a monthly resolution, since generally the 3D outputs provided by chemistry-climate models for multi-model comparisons on multi-decadal timescales are archived as monthly means. This provides a new IAGOS data set (IAGOS-DM) mapped onto the model's grid and time resolution.*

**P1L10 mapped at→mapped onto, or do you mean sampled at?**
**P1L10 by MOCAGE CTM...of CCMI phase-1 → by the MOCAGE CTM...of the CCMI phase-1**
**P1L14 'Good correlations' → 'Good agreement', you don't calculate correlations…**
We confirm 'Good correlations' as a right description, because we did calculate 3D correlations (reported in the ex-Tables 1 and 2, now merged into Table 1).

**P2L4 'only due to...' → 'arising from...'**
**P2L10 'on the processes' → 'in the processes'**
**P2L12 'the main goal [...] lies on the reduction...' → 'the main goal [...] is the reduction...'**
**P2L12 since you indicate the projects of ACCMIP and CCMVal, you should also do so for CCMI, which is IGAC/SPARC. Same on P5 Section 2.2 on CCMI.**
In section 2.2, the following sentence is now starting the paragraph:

*The CCMI project is a common initiative from the IGAC and SPARC programs. The CCMI phase-1 gathers a community of […].*

**P2L18/19 How is it achieved? Recommend adding in a short explainer for 'SD' being 'nudged to observed meteorology' or 'specified dynamics' already here.**

Thanks for this advice. A sentence has been added, as shown below:

*[…] the REF-C1SD experiment aims at assessing the ability of the models to reproduce the actual atmospheric composition for the recent climate time period. For this purpose, a part of its protocol consists in nudging the meteorological fields to meteorological reanalyses based on observations, as indicated by the SD suffix (which stands for "specified dynamics").*

**P4L14 that studied → who studied**

**P4L15 what do you mean by quasi-totality?**
A more precise formulation has been substituted to the previous one. We now say *"the full IAGOS data set corresponding to the cruise phase of flights"*.

**P6L19 I don't understand what the eccentricity of measurements refers to, please clarify.**
The formulation has been clarified, replaced by "*the distance of the measurements from the center within one given grid cell*".

**P6L22 Not clear what this refers to, I assume this approach?**
Exactly. It is specified now.

**P6L25 of MOCAGE simulation → of the MOCAGE simulation**

**P8L15-17 clarify sentence, did you mean...has a negligible impact on the distribution of the IAGOS data from the cruise altitude onto the model vertical grid?**
Exactly. Thanks for this reformulation, now integrated in the text.

**P9L1 in this vicinity → in its vicinity to clarify that you refer back to the grid point.**

**P9 equation 3: the multiplication sign looks formatted in a funny way, suggest to remove.**

**P9L8 suggest to reorder sentence for clarification to → ...on the MOCAGE vertical levels considered which span level 28 to 22 and correspond to…**

**P9L11 Do you really mean to say that all observations on levels 27 and 28 correspond to ascent or descents of the aircraft?**
No. We thus added the word "generally" to avoid any confusion.

**P9L15 This is only true if the observations going into a grid box of your IAGOS-DM is representative for the monthly mean. This will depend on the number of measurements (your N) and the temporal sampling (distributed evenly over the month versus other sampling).**
It is true that we have to be cautious on this formulation. We thus smoothed this sentence, with the following modification (and with the removal of the "month-by-month" expression, that we do not do in this study).
*Thus, IAGOS-DM and MOCAGE-M data sets are spatially consistent and can be used to make gridpoint-to-gridpoint comparisons on climatological timescales, as long as we assume the gridded IAGOS data to be representative of the measurement period.*

**P10L12 inconvenient → inconvenience**

**P18L16 be more specific for the general reader → tropopause transition layer**

**P20L27-29 inter-regional averages is not a valid expression. Cross-regional perhaps...but maybe better an average calculated over all regions…**

**P20L28 as they are similar with zonal averages… → as they are similar to the zonal averages…**

---

## Author Response (AR2)

We are thankful to the modifications proposed by Reviewer 2. We addressed the comments in the same way as in the first authors' response.

Review of revised version of Cohen et al. (2021), Interpol-IAGOS: a new method...

The author response to the comment from Anonymous Referee #1 regarding the use of Specified Dynamics (SD) simulations ( '... the problem with the SD-simulations could be that certain nudging methodologies may lead to introducing noise (and hence too much mixing) especially visible around the tropopause (see discussion in Orbe et al. 2020).') misses the mark a bit. I agree that MOCAGE, being a CTM, would not suffer from problems due to nudging of dynamical variables, however the problems introduced by nudging in a SD simulation should be seen as a caveat to the results found comparing SD simulations and not as a possible application. As argued by Orbe et al (2020), the dynamical noise and resulting loss of consistency between tracers and dynamical variables is a problem with SD simulations and should be pointed out here as a possible aggravating factor. The text inserted at Page 27, Lines 9 – 10 does not adequately bring this forward. It is an important point since the use of SD simulations (or a CTM using reanalysis) is an important component of the current comparison with IAGOS observations as one would expect any significant biases in the position of the tropopause in the model would be reduced making the comparison with IAGOS-DM more straightforward.

We corrected this formulation, as follows:

*To a greater extent, it can be used on a wide range of long-term simulations including* **both** *CCMs* **nudged and** *free runs in order to perform climatological comparisons.* *Precaution must be taken while extending this work to the specified-dynamics simulations from CCMs, regarding the loss of consistency between chemical and dynamical variables that is introduced by nudging, as highlighted in Orbe et al. (2020). Notably, inconsistencies between ozone and potential vorticity are likely to introduce noise in the simulated upper-tropospheric and the lower-stratospheric behaviours.*

Page 4, Lines 21 – 23: There is revised text here that reads 'To compare the REF-C1SD simulations against IAGOS data, interpolating the simulation outputs onto the high-resolution observations would be expensive computationally, and not required because our study is not focused on processes but on climatologies.' and was added to address the comment of Referee #1 on the original text at P4L3, that the interpolation of the model outputs on to the IAGOS observations is 'not possible'. There is new text inserted at Page 4, Lines 16 – 18 that much better addresses the substance of the original comment by Reviewer #1. And even with the focus on climatologies ('... our study is not focused on processes but on climatologies.') would it not be advantageous to have high frequency model outputs that could be interpolated on to the IAGOS observations to construct a climatology that is more directly comparable with the climatology derived from IAGOS? It is difficult to argue against the view that the correct way to perform the comparison would be to have high frequency model outputs and, while computationally expensive, it is nowhere near as computationally expensive as the original model simulations. But for now we do not have high frequency model outputs from multi-model intercomparisons such as CCMI. I would suggest the authors revise the text at Page 4, Lines 21 – 23.

Thanks for this suggestion. We substituted Referee 2's argument to the ones we were using. The new sentence is:

*To compare the REF-C1SD simulations against IAGOS data, interpolating the simulation outputs onto the high-resolution observations would be* **the most accurate way, but high-frequency outputs from multi-model intercomparisons such as CCMI are not available yet.**

**Page 10, Lines 6 – 12: Here the authors have added a brief description of the comparison of sampling MOCAGE daily output with the monthly sampling in response to the second half of the comment of Referee #1 to P4L3 that begins 'In fact, it would be important to prove for a methodology paper as you have presented here that your claim of the gridded IAGOS data being representative of the monthly mean is true.' In the response to the comment, the authors did perform a new analysis that is described in the response to Reviewer #1 using MOCAGE-M-day. There are some statistics from this comparison in the response to Reviewer #1, but the authors do not include any quantitative results from the comparison to MOCAGE-M-day in the article – as it is, the authors state 'MOCAGE-M monthly means could be considered as representative of the month' with no supporting information. In particular, I am not too worried about the sampling in locations with a high frequency of aircraft sampling (the North Atlantic flight corridor, for example) but some mention of the magnitude of the larger differences would be instructive.**

We agree with this comment, the mentioned sentence in the paper should rely on some values describing the distribution of the relative biases between MOCAGE-M-day and MOCAGE-M monthly means. Below is a table showing three percentiles for each species and season, characterizing the relative bias, in absolute values (the same as involved in the FGE equation). We chose to show the percentiles 90, 95 and 99 to the reviewers in order to give more details, but for a better clarity, the manuscript does not include the percentile 95.

| Percentiles | | 90 | 95 | 99 |
|---|---|---|---|---|
| O3 | DJF | 0.104 | 0.136 | 0.228 |
| | MAM | 0.098 | 0.133 | 0.222 |
| | JJA | 0.064 | 0.087 | 0.147 |
| | SON | 0.081 | 0.105 | 0.167 |
| | ANN | 0.060 | 0.080 | 0.131 |
| CO | DJF | 0.079 | 0.109 | 0.181 |
| | MAM | 0.064 | 0.088 | 0.158 |
| | JJA | 0.049 | 0.067 | 0.130 |
| | SON | 0.063 | 0.085 | 0.144 |
| | ANN | 0.041 | 0.055 | 0.103 |

The 90 percentile of the bias spreads from 4.1 to 10.4 %. The 95 percentile is generally lesser than 10 % also, except during boreal winter and spring when the ozone bias reaches until 13.6 %. Concerning the extremely high values, the annual 99 percentile equals 13.1 % for ozone and 10.3 % for CO. For the seasonal biases, the 99 percentile spreads from 14.7 up to 22.8 % for ozone, and from 13.0 up to 18.1 % for CO.

We also calculated PDFs for seasonal cycles, as in the four histograms shown below:

[Figure]

Figure AR1: Histogram showing the PDF of the normalized biases for ozone in the UT, the sample being the relative biases between the seasonal cycles from MOCAGE-M-day and MOCAGE-M during 2003 - 2007, concatenated across the regions. The sample is made of 81 values.

[Figure]

Figure AR2: Same as Fig. AR1 for ozone in the LS. The sample is made of 96 values.

[Figure]

Figure AR3: Same as Fig. AR1 for CO in the UT. The sample is made of 79 values.

[Figure]

Figure AR4: Same as Fig. AR1 for CO in the LS. The sample is made of 96 values.

These histograms show that the quasi-totality of the biases are below 5 %. Ozone biases outreach 10 % twice in the UT (-10.7 % in Northwest America, in October; 15.2 % in Middle East, in February) and four times in the LS (-10.9 % in Northwest America, in October; -13.2 % in North Atlantic, in November; -14.1 and 10.4 % in the West Mediterranean basin, in January and September respectively), the maximum value being ~ 15 %.

The changes in the paper are as follows:
- At the end of Section 3.2:

*The latter point has been tested (not shown) on a 5-year subsample using the simulation daily outputs instead of monthly outputs. It consisted of comparing MOCAGE-M to a test product derived by calculating monthly averages from the daily outputs and applying a mask based on the IAGOS daily sampling. **The results from this test are briefly presented in Sect. 4.***

- In the results section, a short subsection has been added:

*4.1 Monthly representativeness*

*A first step in the assessment of the methodology consists of testing the monthly representativeness of the IAGOS-DM mean values, in order to evaluate the temporal consistency between IAGOS-DM and MOCAGE-M. For this purpose, as mentioned at the end of Section 3.2, we compared MOCAGE-M to a test product derived by calculating monthly averages from the simulation daily outputs, after applying a mask based on the IAGOS daily sampling. For this test, the chosen period spreads from 2003 until 2007 included, an uninterrupted measurement period for both ozone and CO. Concerning the mean 3D distributions, a mean normalized difference between the two products has been found below 1.7 % for each season and each species. In absolute values, 10 % of the yearly mean biases are greater than 6.0 % (4.1 %) for ozone (CO), and 1 % greater than 13.1 % (10.3 %). Seasonal mean biases are characterized by a 90 percentile generally lower than 10 %, and a 99 percentile from 14.7 up to 22.8 % for ozone and from 13.0 up to 18.1 % for CO. The maximum values correspond to winter and spring. Concerning the seasonal cycles, the relative difference between the two MOCAGE products was found to be almost systematically below 5 %, and amongst all the regions, its ozone values seldom outreach 10 %, with a maximum value at 15.2 %. In conclusion of this comparison, the similar results obtained between MOCAGE-M and the test product suggested that in most cases, the IAGOS-DM monthly means could be considered as representative of the month.*

**On the reply of the authors to the comment 'On a similar note, it would also be good to see what the benefit of the weighted gridding versus a gridding of the observations without the weighting function would be...' To estimate the benefit I would think you would have the weighted and unweighted gridding and compare these two products to some estimate of the truth. Here, we have the original aircraft IAGOS observations, a weighted gridding of the IAGOS observations, an unweighted gridding of the IAGOS observations, and the model output. One could show differences between the weighted and unweighted gridding, but how does one show the benefits of weighted gridding? To put it another way, because the model differences are smaller for weighted or unweighted gridding, does that mean one is more correct? I suggest the authors be careful about stating they have analyzed the benefits of weighting. In particular, I am not convinced the authors have shown that 'using a weighting function is a necessary step for a more accurate assessment' as stated in the abstract of the revised version at Page 2, Lines 8 – 11. Given the difficulty of showing a benefit to weighting, I would suggest the authors revise the text to be clear they are showing 'differences' and not 'benefits'.**

It is true that we did not choose our words correctly for this test (MOCAGE-M vs IAGOS-DM against MOCAGE-M-noW vs IAGOS-DM-noW), since we cannot show that the results are "better" although the scores are enhanced.

In the abstract:
*Along this model evaluation, we also assess the* **differences caused** *by the use of a weighting function in the method when projecting the IAGOS data onto the model grid compared to the scores derived in a simplified way. We conclude that the data projection onto the model's grid allows to filter out biases arising from either spatial or temporal resolution*, **and the use of a weighting function yields different results, here by enhancing the assessment scores**. *Beyond the MOCAGE REF-C1SD evaluation presented in this paper, the method could be used by CCMI models for individual assessments in the UTLS and for model intercomparisons with respect to the IAGOS data set.*

And in the last sentence in Section 4.1:

*The general improvement of normalised biases, normalised errors and spatial correlations, compared to a simplified gridding method,* **suggests** *that the use of a weighting function in our methodology* **can significantly enhance the model assessment.**

**Page 20, Lines 24 – 27: The text at this point was added in response to a comment from Reviewer 2 about the lack of directly addressing the effects of time averaging for the comparison with IAGOS data around the tropopause ('It became a bit confusing when the discussion of the comparison of IAGOS-HR and IAGOS-DM zeroed in on the effect of mis-classification of points in either the UT or LS (see the comment on Page 19, Lines 4 – 6) and ignored the effect of time averaging.'). The text in this section still discusses the inability to correctly classify air masses using monthly average PV, but the problem is much more fundamental than. There is no clean separation of the stratosphere and troposphere left in the IAGOS-DM data after it is monthly averaged on constant pressure surfaces. By using monthly averages, of either the IAGOS data on a particular pressure level (IAGOS-DM) or model data, the sharp separation of tropospheric and stratospheric air around the tropopause, which would be preserved in the IAGOS-HR data, is lost. This will be a fundamental problem with analysing monthly average data in the vicinity of the tropopause and is illustrated to some extent by the differences between IAGOS-HR and IAGOS-DM. This is the point that is still missing in the text and, I think, it is an important one because it illustrates that treating the IAGOS-DM data in the way it is treated does make it more like the monthly average model data so the comparison of IAGOS-DM and the model is more valid. But because it is monthly averaged data the sharpness of the separation between tropospheric and stratospheric air masses is lost to some extent.**

As written in the first authors' response, we agree with this explanation. We thought our previous answer to this comment was sufficient, as we consider that the misclassification of individual measurement points and the loss of sharpness in the tropopause are the same direct consequence of the loss of resolution caused by time averaging, because both generate a mixing between the UT and the LS. In other words, we thought we were telling the same thing, although with a different view. In order to make it explicit, now we have modified the text as follows:
*In other words, the effect of time averaging leads to* **a loss of tropopause sharpness, thus resulting in** *a mis-classification of a non-negligible part of the individual measurements. For a given layer, it introduces a bias due to unexpected mixing with another layer.*

**A couple of minor comments:**

**Page 6, Line 12: 'expanding from 1980 to 2010' should be 'extending from 1980 to 2010'**
Thanks for this correction, the change has been made.

**Page 10, Lines 27 – 28: The new text 'less measurements are needed to characterize the climatologies' in reference to the CO observations implies that the CO climatology is somehow equally well characterized as the ozone climatology using less data. It would seem that the reason 60 CO observations are judged sufficient is that the sampling period for CO is shorter and to use the same Nthres=100 as for ozone means throwing out too much data. It is not that less measurements are needed to characterize the the CO climatology, so the wording should be modified here.**

We agree with this comment. We reworded the text, as quoted below:
*Accounting for the shorter CO measurement period compared to $O_3$ (~60 % of the $O_3$ period), the same threshold applied to the CO climatologies would result in a greater proportion of filtered-out grid cells. Thus, the corresponding $N_{thres}$ threshold for this species is derived by applying a factor 0.6, leading to 60.*